# Mixture-of-Experts Meets In-Context Reinforcement Learning

**Wenhao Wu**[1]  **Fuhong Liu**[1]  **Haoru Li**[1]  **Zican Hu**[1]
**Daoyi Dong**[2]  **Chunlin Chen**[1]  **Zhi Wang**[1*]

[1]Nanjing University    [2]University of Technology Sydney
{wenhaowu, chika, haoruli, zicanhu}@smail.nju.edu.cn
daoyi.dong@uts.edu.au   {clchen, zhiwang}@nju.edu.cn

## Abstract

In-context reinforcement learning (ICRL) has emerged as a promising paradigm for adapting RL agents to downstream tasks through prompt conditioning. However, two notable challenges remain in fully harnessing in-context learning within RL domains: the intrinsic multi-modality of the state-action-reward data and the diverse, heterogeneous nature of decision tasks. To tackle these challenges, we propose **T2MIR** (**T**oken- and **T**ask-wise **M**oE for **I**n-context **R**L), an innovative framework that introduces architectural advances of mixture-of-experts (MoE) into transformer-based decision models. T2MIR substitutes the feedforward layer with two parallel layers: a token-wise MoE that captures distinct semantics of input tokens across multiple modalities, and a task-wise MoE that routes diverse tasks to specialized experts for managing a broad task distribution with alleviated gradient conflicts. To enhance task-wise routing, we introduce a contrastive learning method that maximizes the mutual information between the task and its router representation, enabling more precise capture of task-relevant information. The outputs of two MoE components are concatenated and fed into the next layer. Comprehensive experiments show that T2MIR significantly facilitates in-context learning capacity and outperforms various types of baselines. We bring the potential and promise of MoE to ICRL, offering a simple and scalable architectural enhancement to advance ICRL one step closer toward achievements in language and vision communities. Our code is available at https://github.com/NJU-RL/T2MIR.

## 1 Introduction

Reinforcement learning (RL) is emerging as a powerful mechanism for training autonomous agents to solve complex tasks in interactive environments [1, 2], unleashing its potential across frontier challenges including preference optimization [3], training diffusion models [4], and reasoning [5, 6, 7] such as OpenAI-o1 [8] and DeepSeek-R1 [9]. Recent studies have been actively exploring how to harness the in-context learning capabilities of the transformer architecture to achieve substantial improvements in RL's adaptability to downstream tasks through prompt conditioning without any model updates, i.e., in-context RL (ICRL) [10]. Current research in offline settings encompasses two primary branches, algorithm distillation (AD) [11] and decision-pretrained transformer (DPT) [12], owing to their simplicity and generality. They share a common structure that uses across-episodic transitions as few-shot prompts to a transformer policy, and following-up studies continue to increase the in-context learning capacity by hierarchical decomposition [13], noise distillation [14], model-based planning [15], dynamic programming updates [16, 17], etc. [18, 19, 20].

---

[*]Correspondence to Zhi Wang <zhiwang@nju.edu.cn>.

39th Conference on Neural Information Processing Systems (NeurIPS 2025).

Despite these efforts, two notable challenges remain in fully harnessing in-context learning within decision domains, as RL is notably more dynamic and complex than supervised learning. The first is the intrinsic multi-modality of datasets. In language or vision communities, the inputs to transformers are atomic words or pixels with consistent semantics in the representation space [21]. In ICRL, the prompt inputs are typically transition samples containing three modalities of state, action, and reward with large semantic discrepancies. States like physical quantities are usually continuous in nature in RL, actions like joint torques tend to be more high-frequency and less smooth, and rewards are simple scalars that can be sparse over long sequences [22]. The second is task diversity and heterogeneity. Compared to supervised learning, RL models are more prone to overfit the training set and struggle with generalization across diverse tasks [23]. The learning efficiency can be hindered by intrinsic gradient conflicts in challenging scenarios where tasks vary significantly [24]. [2] The above limitations raise a key question: *Can we design a scalable ICRL framework to tackle the multi-modality and diversified task distribution within a single transformer, advancing RL's in-context learning capacities one step closer to achievements in language and vision communities?*

In the era of large language models (LLMs), the mixture-of-experts (MoE) architecture [25] has shown remarkable scaling properties and high transferability while managing computational costs, such as in Gemini [26], Llama-MoE [27], and DeepSeekMoE [28]. The architectural advancement also extends to various domains such as computer vision [29], image generation [30], and RL [31, 32], highlighting the significant potential and promise of MoE models. Intuitively, MoE architectures can serve as an encouraging remedy to tackle the two aforementioned bottlenecks in ICRL. First, MoE enables different experts to process tokens with distinct semantics more effectively [33]. This is a natural match for processing the multiple modalities within the state-action-reward sequence. Second, MoE can alleviate gradient conflicts by dynamically allocating gradients to specialized experts for each input through the sparse routing mechanism [34]. This property is promising for handling a broad task distribution with significant diversity and heterogeneous complexity.

Inspired by this, we propose an innovative framework **T2MIR** (**T**oken- and **T**ask-wise **M**oE for **I**n-context **RL**), which for the first time harnesses architectural advances of MoE to develop more scalable and competent ICRL algorithms. We substitute the feedforward layer in transformer blocks with two parallel ones: a token-wise MoE and a task-wise MoE. First, the token-wise MoE layer is responsible for automatically capturing distinct semantic features of input tokens within the multimodal state-action-reward sequence. We include a load-balancing loss and an importance loss to avoid tokens from all modalities collapsing onto minority experts. Second, the task-wise MoE layer is designed to assign diverse tasks to specialized experts with sparse routing, effectively managing a broad task distribution while alleviating gradient conflicts. To facilitate task-wise routing, we introduce a contrastive learning method to maximize the mutual information between the task and its router representation in the MoE layer, enabling more precise capture of task-relevant information. Finally, the outputs of the two parallel MoE components are concatenated and fed into the next layer.

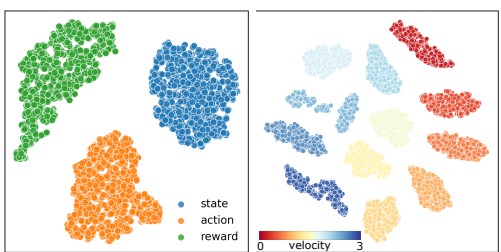

Figure 1: t-SNE visualization of expert assignments on Cheetah-Vel where tasks differ in target velocities. Left: token-wise MoE enables different experts to process tokens with distinct semantics. Right: task-wise MoE effectively manages a broad task distribution, where the difference between expert assignments is positively related to the difference between tasks.

In summary, our main contributions are threefold:

- We unleash RL's in-context learning capacities with a simple and scalable architectural enhancement. To our knowledge, we are the first to bring the potential and promise of MoE to ICRL.

- We design a token-wise MoE to facilitate processing the distinct semantics within multi-modal inputs, and a task-wise MoE to tackle a diversified task distribution with reduced gradient conflicts.

- We build our method upon AD and DPT, and conduct extensive experiments on various benchmarks to show superiority over competitive baselines and visualize deep insights into performance gains.

---

[2] For example, given two navigation tasks where the goals are in adverse directions, the single RL model ought to execute completely opposite decisions under the same states for the two tasks.

## 2 Related Work

**In-Context RL.** Tackling task generalization is a long-standing challenge in RL. Early methods address this via the lens of meta-RL, including the memory-based (e.g., $RL^2$ [35] and LLIRL [36, 37]), the optimization-based (e.g., MAML [38] and MACAW [39]), and the context-based (e.g., VariBAD [40], Meta-DT [41], etc. [42, 43]). With the emergence of transformers that show remarkable in-context learning abilities [44], the community has been exploring its potential to enhance RL's generalization via prompt conditioning without model updates [10].

Many ICRL methods have emerged, each differing in how to train and organize the context. In online settings, AMAGO [45, 46] trains long-sequence transformers over entire rollouts with actor-critic learning, and [47] leverages the S4 (structured state space sequence) model to handle long-range sequences for ICRL tasks. Classical studies in offline settings include AD [11] and DPT [12]. AD trains a causal transformer to predict actions given preceding histories as context, and DPT predicts the optimal action given a query state and a prompt of interactions. Follow-up studies enhance in-context learning from various algorithmic aspects [14, 18, 19, 20, 16, 17], such as IDT with hierarchical structure [13] and DICP with model-based planning [15]. Complementary to these *algorithmic* studies, we introduce MoE to advance ICRL's development from an *architectural* perspective.

**Mixture-of-Experts.** The concept of MoE is first proposed in [48], which employs different expert networks plus a gating network that decides which experts should be used for each training case [49]. Then, it is applied to tasks of language modeling and machine translation [25], gradually showing prominent performance in scaling up transformer model size while managing computational costs [50, 51]. Up to now, MoE architectures have become a standard component for advanced LLMs, such as Gemini [26], Llama-MoE [27], and DeepSeekMoE [28]. Recent efforts have explored extending MoE advancements to RL domains. [31] incorporates soft MoE modules into value-based RL agents, and performance improvements scale with the number of experts used. [32] uses an MoE backbone with a CNN encoder to process visual inputs and improves the policy learning ability to handle complex robotic environments. [34] strengthens decision transformers with MoE to reduce task load on parameter subsets, with a three-stage training mechanism to stabilize MoE training. Accompanied by these encouraging efforts, we aim to harness the potential of MoE for ICRL.

## 3 Preliminaries

### 3.1 In-Context Reinforcement Learning

We consider a multi-task offline RL setting, where tasks follow a distribution $M_n = \langle \mathcal{S}, \mathcal{A}, \mathcal{T}_n, \mathcal{R}_n, \gamma \rangle \sim P(M)$. Tasks share the same state space $\mathcal{S}$ and action space $\mathcal{A}$, while differing in the reward function $\mathcal{R}$ or transition dynamics $\mathcal{T}$. An offline dataset $\mathcal{D}^n = \{(s_j^n, a_j^n, r_j^n, s_j^{'n})\}_{j=1}^J$ is collected by arbitrary behavior policies for each task out of a total of $N$ training tasks. The agent can only access the offline datasets $\{\mathcal{D}^n\}_{n=1}^N$ to train an in-context policy as $\pi\left(a^n | s^n; \tau_{\text{pro}}^n\right)$, where $\tau_{\text{pro}}^n$ is a trajectory prompt that encodes task-relevant information. At test time, the trained policy is evaluated on unseen tasks sampled from $P(M)$ by directly interacting with the environment. The prompt is initially empty and gradually constructed from history interactions. The objective is to learn an in-context policy to maximize the expected episodic return over test tasks as $J(\pi) = \mathbb{E}_{M \sim P(M)}[J_M(\pi)]$.

### 3.2 Mixture-of-Experts

A standard sparse MoE layer consists of $K$ experts $\{E_1, ..., E_K\}$ and a router $G$. The router predicts an assignment distribution over the experts given the input $x$. Following the common practice [25, 32], we only activate the top-$k$ experts to process the inputs. In general, the number of activated experts is fixed and much smaller than the total number of experts, thus scaling up model parameters and significantly reducing computational cost. Formally, the output of the MoE layer can be written as

$$w(i; x) = \text{softmax}\left(\text{topk}(G(x))\right)[i], \quad y = \sum_{i=1}^K w(i; x) E_i(x), \tag{1}$$

where $\text{topk}(\cdot)$ selects top $k$ experts based on the router output $G(x)$. $\text{softmax}(\cdot)$ normalizes top-$k$ values into a weight distribution $w(\cdot)$. Probabilities of non-selected experts are set to 0.

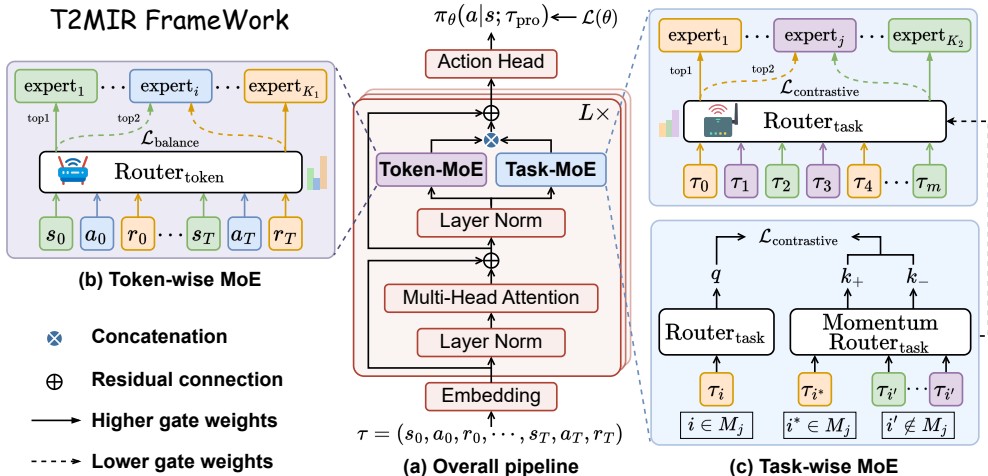

Figure 2: The overview of T2MIR. (a) **Overall pipeline**: we substitute the feedforward layer in causal transformer blocks with two parallel MoE layers and concatenate their outputs to feed into the next layer. (b) **Token-wise MoE**: it automatically captures distinct semantic features within the multi-modal $(s, a, r)$ inputs, and uses $\mathcal{L}_{\text{balance}}$ as regularization loss to avoid tokens from all modalities collapsing onto minority experts. (c) **Task-wise MoE**: it assigns diverse tasks to specialized experts, and includes a contrastive learning loss $\mathcal{L}_{\text{contrastive}}$ to enhance task-wise routing via more precise capture of task-relevant information, where $\tau_i$ is the query and $\tau_{i^*}/\tau_{i'}$ are positive/negative keys.

## 4 Method

In this section, we present **T2MIR** (**T**oken- and **T**ask-wise **M**oE for **I**n-context **R**L), an innovative framework that harnesses architectural advancements of MoE to tackle ICRL challenges of the multi-modality and diversified task distribution. Figure 2 illustrates the method overview, followed by detailed implementations. Algorithm pseudocodes are presented in Appendix A.

### 4.1 Token-wise MoE

By interacting with the outer environment, we gather the data represented as a sequence of state-action-reward transitions in the form of $\tau = (s_0, a_0, r_0, ..., s_T, a_T, r_T)$, which often serves as the prompt input to ICRL models. The state is usually continuous in nature, which can contain physical quantities of the agent (e.g., position, velocity, and acceleration) or an image in visual RL [32]. The action tends to be more high-frequency and less smooth, such as joint torques or discrete commands. The reward is a simple scalar that is often sparse over long horizons. In language or vision communities, the inputs to transformers are atomic words or pixels with consistent semantics in the representation space [21]. In contrast, ICRL encounters a new challenge of processing the prompt data that encompasses three modalities with significant semantic discrepancies.

Inspired by the surprising discoveries that the MoE structures can effectively capture distinct semantic features of input tokens [50, 33], we introduce a token-wise MoE layer to tackle the multiple modalities within the state-action-reward sequence. As shown in Figure 2-(b), each element in the multi-modal sequence corresponds to a token. The MoE consists of $K_1$ experts $E_{\text{tok}}$ and a router $G_{\text{tok}}$, and the router learns to assign each token to specific experts with noisy top-$k$ gating. Let $h$ denote the hidden state of a given token after self-attention calculation. The token router $G_{\text{tok}}$ computes a distribution of logits $\{G_{\text{tok}}(i|h)\}_{i=1}^{K_1}$ for expert selection. The top-$k$ experts will be selected to process this token, and their outputs will be weighted to produce the final output $y_{\text{tok}}$ as

$$w_{\text{tok}}(i; h) = \text{softmax}\left(\text{topk}(G_{\text{tok}}(i|h))\right)[i], \qquad y_{\text{tok}} = \sum_{i=1}^{K_1} w_{\text{tok}}(i; h) \cdot E_{\text{tok}}(i|h). \qquad (2)$$

The router tends to converge on producing large weights for the same few experts, as the favored experts are trained more rapidly and thus are selected even more by the router. In order to avoid

tokens from all modalities collapsing onto minority experts, we incorporate a regularization term to balance expert utilization that consists of an importance loss and a load-balancing loss [25] as

$$\mathcal{L}_{\text{balance}} = w_{\text{imp}} \cdot CV(\text{Imp}(h))^2 + w_{\text{load}} \cdot CV(\text{Load}(h))^2, \tag{3}$$

where $CV(\cdot)$ denotes the coefficient of variation, and $w_{\text{imp}}$ and $w_{\text{load}}$ are hand-tuned scaling factors. The first item encourages all experts to have equal importance as defined by $\text{Imp}(h) = \sum_h G_{\text{tok}}(\cdot|h)$. The second item encourages experts to receive roughly equal numbers of training examples, where $\text{Load}(\cdot)$ is a smooth estimator of the number of examples assigned to each expert for a batch of inputs. A more detailed description of the balance loss can be found in Appendix B.

## 4.2  Task-wise MoE

In typical ICRL settings, a single policy model is trained across multiple tasks. The learning efficiency can be impeded by intrinsic gradient conflicts in challenging scenarios with significant task variation. For instance, given two navigation tasks where the goals are in adverse directions, the policy ought to make contrary decisions under the same states. Given the same state-action trajectories, the two tasks can produce exactly opposite policy gradients, leading to a sum of zero gradient during training.

The MoE structure was originally proposed to use different parts of a model, called experts, to specialize in different tasks or different aspects of a task [48, 49]. Drawing from this natural inspiration, we introduce a task-wise MoE layer to handle a broad task distribution with significant diversity and heterogeneous complexity, leveraging modular expert learning to alleviate gradient conflicts. As shown in Figure 2-(c), the MoE consists of $K_2$ experts $E_{\text{task}}$ and a router $G_{\text{task}}$, and the router learns to assign a sequence of tokens to specialized experts *at the task level*. Given a trajectory $\tau = (s_0, a_0, r_0, ..., s_T, a_T, r_T)$ from some task $M$, we use $\bar{h}$ to denote the average hidden state of all three-modality tokens after self-attention calculation. The task router $G_{\text{task}}$ computes a distribution of logits $\{G_{\text{task}}(i|\bar{h})\}_{i=1}^{K_2}$ for expert selection. The top-$k$ experts will be selected to process these tokens, and their outputs will be weighted to produce the final output $y_{\text{task}}$ as

$$w_{\text{task}}(i; \bar{h}) = \text{softmax}\left(\text{topk}(G_{\text{task}}(i|\bar{h}))\right)[i], \qquad y_{\text{task}} = \sum_{i=1}^{K_2} w_{\text{task}}(i; \bar{h}) \cdot E_{\text{task}}(i|\bar{h}). \tag{4}$$

**Task-wise Routing via Contrastive Learning.** The task-wise router can be viewed as an encoder for extracting task representations from input tokens, based on the intuition that similar tasks are preferred to the same expert and tasks with notable differences are allocated to disparate experts. An ideally discriminative task representation should accurately capture task-relevant information from offline datasets, while remaining invariant to behavior policies or other unrelated factors. To achieve this goal, we propose to maximize the mutual information between the task and its router representation, enhancing task-wise routing that minimally preserves task-irrelevant information.

Formally, the mutual information $I(\cdot)$ aims to quantify the uncertainty reduction of one random variable when the other is observed, measuring their mutual dependency. Mathematically, we formalize the router $G_{\text{task}}$ as a probabilistic encoder $z \sim G_{\text{task}}(\cdot|\bar{h})$, where $z$ denotes the task representation. The $\bar{h}$ distribution is determined by its task $M$, where $M \sim P(M)$. The objective for the router is:

$$\max I(z; M) = \mathbb{E}_{z,M}\left[\log \frac{p(M|z)}{p(M)}\right]. \tag{5}$$

Drawing inspiration from noise contrastive estimation (InfoNCE) [52] in the contrastive learning literature [53, 54], we derive a lower bound of Eq. (5) using the following theorem.

**Theorem 1.** *Let $\mathcal{M}$ denote a set of tasks following the task distribution $P(M)$, and $|\mathcal{M}| = N$. $M \in \mathcal{M}$ is a given task. Let $\bar{h} = f(\tau)$, $z \sim G_{task}(\cdot|\bar{h})$, and $e(\bar{h}, z) = \frac{p(z|\bar{h})}{p(z)}$, where $\tau$ is a trajectory from task $M$ and $\bar{h}$ is average hidden state of all tokens after the self-attention calculation function $f(\cdot)$. Let $\bar{h}'$ denote the average hidden state generated by any task $M' \in \mathcal{M}$, then we have*

$$I(z; M) \geq \log N + \mathbb{E}_{\mathcal{M},z,\bar{h}}\left[\log \frac{e(\bar{h}, z)}{\sum_{M' \in \mathcal{M}} e(\bar{h}', z)}\right]. \tag{6}$$

Appendix C presents the complete proof. Since direct evaluation of $p(z)$ or $p(z|\bar{h})$ is intractable, we employ NCE and importance sampling techniques that compare the target value with randomly

sampled negative ones. Consequently, we approximate $e$ using the exponential of a score function, a similarity metric between the latent codes of two examples. Given a batch of $m$ trajectories $(\tau_1, ..., \tau_m)$, we denote router representation $z_i$ from $\tau_i$ as the query $q$, and representations of other trajectories as the keys $\mathbb{K} = \{z_1, ..., z_m\} \setminus z_i$. Points from the same task with the query, $i \in M_j$, are set as positive key $\{k_+\}$ and those from different tasks are set as negative $\{k_-\} = \mathbb{K} \setminus \{k_+\}$. We adopt bilinear products [53] as the score function, with similarities between the query and keys computed as $q^\top W k$, where $W$ is a learnable parameter matrix. Then, we formulate a sampling-based version of the tractable lower bound as the InfoNCE loss for the contrastive router as

$$\mathcal{L}_{\text{contrastive}} = -\log \frac{\exp(q^\top W k_+)}{\exp(q^\top W k_+) + \exp(q^\top W k_-)} = -\log \frac{\sum_{i^* \in M_j} \exp(z_i^\top W z_{i^*})}{\sum\limits_{i^* \in M_j} \exp(z_i^\top W z_{i^*}) + \sum\limits_{i' \notin M_j} \exp(z_i^\top W z_{i'})} \quad (7)$$

It optimizes an $N$-way classification loss to classify the positive pair among all pairs, equivalent to maximizing a lower bound on mutual information in Eq. (6), with this bound tightening as $N$ gets larger. Following MoCo [55], we maintain a query router $G_{\text{task}}^q$ and a key router $G_{\text{task}}^k$, and use a momentum update with coefficient $\beta \in [0, 1)$ to ensure the consistency of the key representations as

$$G_{\text{task}}^k \leftarrow \beta G_{\text{task}}^k + (1 - \beta) G_{\text{task}}^q, \quad (8)$$

where only query parameters $G_{\text{task}}^q$ are updated through backpropagation.

### 4.3 Scalable Implementations

We develop scalable implementations of T2MIR using two mainstream ICRL backbones that offer promising simplicity and generality: AD [11] and DPT [12], yielding two T2MIR variants as follows.

**T2MIR-AD.** It trains an MoE-augmented causal transformer policy $\pi_\theta$ to autoregressively predict actions given preceding learning histories $his_t$ as the prompt:

$$his_t := (s_0, a_0, r_0, ..., s_{t-1}, a_{t-1}, r_{t-1}, s_t, a_t, r_t) = (s_{\leq t}, a_{\leq t}, r_{\leq t}). \quad (9)$$

During training, we minimize a negative log-likelihood loss over $N$ individual tasks as

$$\mathcal{L}(\theta) = -\sum_{n=1}^N \sum_{t=0}^{T-1} \log \pi_\theta \left( a = a_t^n \mid s_t^n; \ \tau_{\text{pro}}^n \right), \quad \text{where } \tau_{\text{pro}}^n := his_{t-1}^n, \quad (10)$$

which can be derived as a cross-entropy loss for discrete actions, and a mean-square error (MSE) loss when the action space is continuous. Akin to AMAGO [45], we adopt Flash Attention [56] to enable long context lengths on a single GPU. For the context, we use the same position embedding for $s_t$, $a_t$, and $r_t$ to maintain semantics of temporal consistency.

**T2MIR-DPT.** It trains a transformer policy to predict an optimal action given a query state and a prompt of interaction transitions. In practice, we randomly sample a trajectory $\tau_{\text{pro}}$ from the dataset $\mathcal{D}$ as the task prompt for each query-label pair $(s_t, a_t^*)$, and train the policy to minimize the loss as

$$\mathcal{L}(\theta) = -\sum_{n=1}^N \sum_{t=0}^T \log \pi_\theta \left( a = a_t^{n*} \mid s_t^n; \ \tau_{\text{pro}}^n \right), \quad \text{where } \tau_{\text{pro}}^n \sim \mathcal{D}^n, \quad (11)$$

Different from AD in Eq. (10), DPT requires labeling optimal actions for query states to construct the offline dataset $\sum_n \sum_t (s_t^n, a_t^{n*})$, inheriting principles from pure imitation learning.

## 5 Experiments

We comprehensively evaluate our methods on popular benchmarking domains using various qualities of offline datasets. In general, we aim to answer the following research questions:

- Can T2MIR demonstrate consistent superiority on in-context learning capacity when tested in unseen tasks, compared to various strong baselines? (Sec. 5.1)

- How do the token-wise and the task-wise MoE layers affect T2MIR's performance, respectively (Sec. 5.2)? We also gain deep insights into each component via visualizations in Sec. 5.4.

- How robust is T2MIR across diverse settings? We construct offline datasets with varying qualities to extensively evaluate T2MIR and baselines. Also, we investigate T2MIR's sensitivity to various hyperparameters, such as the expert configuration and the loss ratio. (Sec. 5.3 and Appendix G)

**Environments.** We evaluate T2MIR on four benchmarks that are widely used in multi-task settings: i) the discrete environment `DarkRoom`, which is a grid-world environment with multiple goals; ii) the 2D navigation environment `Point-Robot`, which aims to navigate to a target position in a 2D plane; iii) the multi-task MuJoCo locomotion control environment, containing `Cheetah-Vel` and `Walker-Param`; iv) the Meta-World manipulation platform, including `Reach` and `Push`. We construct three datasets with different qualities: `Mixed`, `Medium-Expert`, and `Medium`. Appendix D presents more details about environments and dataset construction.

**Baselines.** We compare `T2MIR` to five competitive baselines, including four ICRL methods AD [11], DPT [12], IDT [13], DICP [15], and a context-based offline meta-RL approach UNICORN [42]. More details about baselines are given in Appendix E. As DICP has three implementations, we use the best results among them in our experiments.

For all experiments, we conduct evaluations across five random seeds. The mean of the obtained return is plotted as the bold line, with 95% bootstrapped confidence intervals of the mean results indicated by the lighter shaded regions. Appendix F gives implementation details of T2MIR-AD and T2MIR-DPT. Appendix G presents comprehensive hyperparameter analysis, including expert selection in token-wise and task-wise MoE, InfoNCE loss ratio, and the positioning of MoE layers.

## 5.1 Main Results

We compare our method against various baselines under an aligned evaluation setting, where prompts or contexts are generated from online environment interactions to infer task beliefs during testing. Figure 3 and Table 1 present test return curves and numerical results of the best performance of T2MIR and baselines using Mixed datasets. In these diverse environments with varying reward functions or transition dynamics, both T2MIR-AD and T2MIR-DPT achieve significantly superior performance regarding the in-context learning speed and final asymptotic results compared to the strong baselines. In most cases, our two implementations take the top two rankings for the best and second-best performance, showcasing comparably strong generalization abilities to unseen tasks. It highlights the effectiveness of introducing MoE architectures to the two mainstream ICRL backbones. Notably, DICP exhibits a faster learning speed and comparable asymptotic performance compared to our T2MIR in DarkRoom. In this environment with a small state-action space, DICP naturally enables more efficient policy search via look-ahead model-based planning. Then, DICP struggles when facing more complicated environments like Reach and Push, as it is more difficult to find high-return trajectories when planning in a large space. In contrast, the superiority of T2MIR is more pronounced in harder problems, underscoring its potential to tackle challenging ICRL tasks.

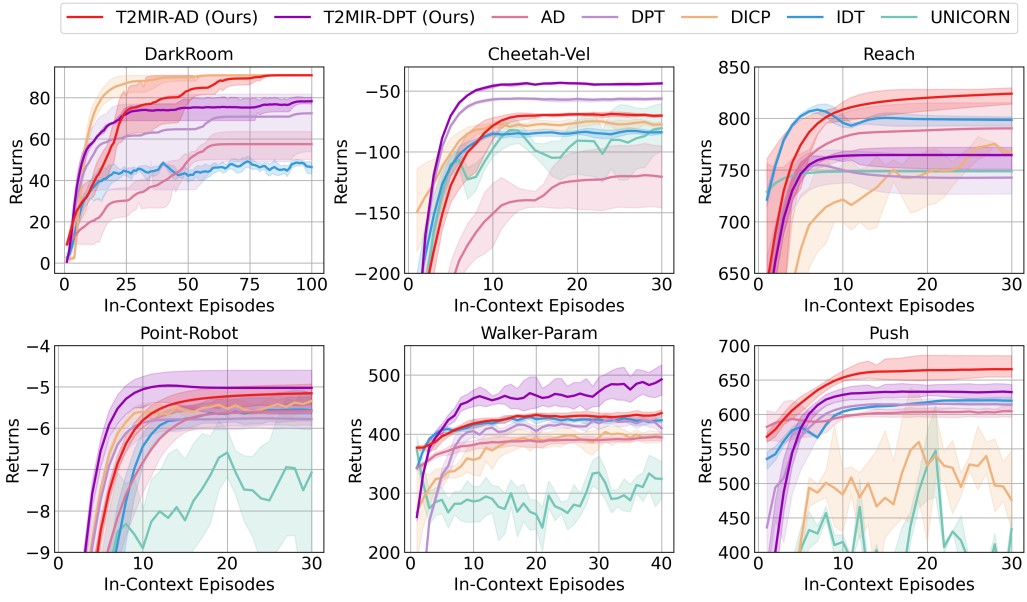

Figure 3: Test return curves of two T2MIR implementations against baselines using Mixed datasets.

Table 1: Test return of T2MIR against baselines using Mixed datasets, i.e., numerical results of the best performance from Figure 3. Best result in **bold** and second best underline.

| Environments | T2MIR-AD | T2MIR-DPT | AD | DPT | DICP | IDT | UNICORN |
|---|---|---|---|---|---|---|---|
| DarkRoom | **90.9**±0.0 | 78.3±1.4 | 57.5±5.6 | 72.5±13.1 | **90.9**±0.0 | 49.2±2.8 | — |
| Point-Robot | −5.2±0.3 | **−5.0**±0.5 | −5.6±0.4 | −5.8±0.3 | −5.3±0.2 | −5.5±0.5 | −6.6±0.6 |
| Cheetah-Vel | −68.9±2.1 | **-43.2**±0.8 | −119.2±30.4 | −56.1±1.0 | −74.9±3.2 | −82.8±3.0 | −80.6±22.8 |
| Walker-Param | 435.7±7.2 | **492.8**±29.3 | 395.2±7.1 | 425.9±9.7 | 403.7±15.4 | 429.4±5.8 | 372.8±24.6 |
| Reach | **823.9**±7.1 | 764.6±9.3 | 790.4±20.2 | 754.9±15.0 | 775.3±1.1 | 808.5±3.6 | 748.8±0.1 |
| Push | **665.8**±13.9 | 633.2±8.6 | 604.7±6.3 | 615.4±6.0 | 560.0±64.2 | 620.6±4.6 | 547.4±67.2 |

## 5.2 Ablation Study

We compare T2MIR to three ablations: 1) `w/o Task-MoE`, it only retains the token-wise MoE layer and regularizes the router with balance loss in Eq. (3); 2) `w/o Token-MoE`, it only retains the task-wise MoE layer and enhances the router with contrastive loss in Eq. (7); and 3) `w/o all`, it degrades to vanilla AD and DPT. We keep the same amount of activated parameters in our T2MIR and three ablations for a fair comparison. Figure 4 and Table 2 present test return curves and numerical results of ablation study on T2MIR-AD and T2MIR-DPT.

First, both T2MIR architectures show decreased performance when any MoE is removed, with the worst occurring when both MoE layers are excluded. It demonstrates that the two kinds of MoE designs are essential for T2MIR's capability and they complement each other. Second, with the AD backbone, ablating the token MoE incurs a more significant performance drop than ablating the task MoE, indicating that the token MoE plays a more vital role for T2MIR-AD. AD takes long training histories as context, which can contain redundant trajectories for identifying the underlying task, akin to the memory-based meta-RL approach RL[2] [35]. Hence, using the token MoE to handle multi-modality within the long-horizon context may yield greater benefits. Third, the situation is reversed with the DPT backbone, as the task MoE is more essential for T2MIR-DPT. DPT takes a small number of transitions as the task prompt, placing greater emphasis on accurately identifying task-relevant information from limited data.

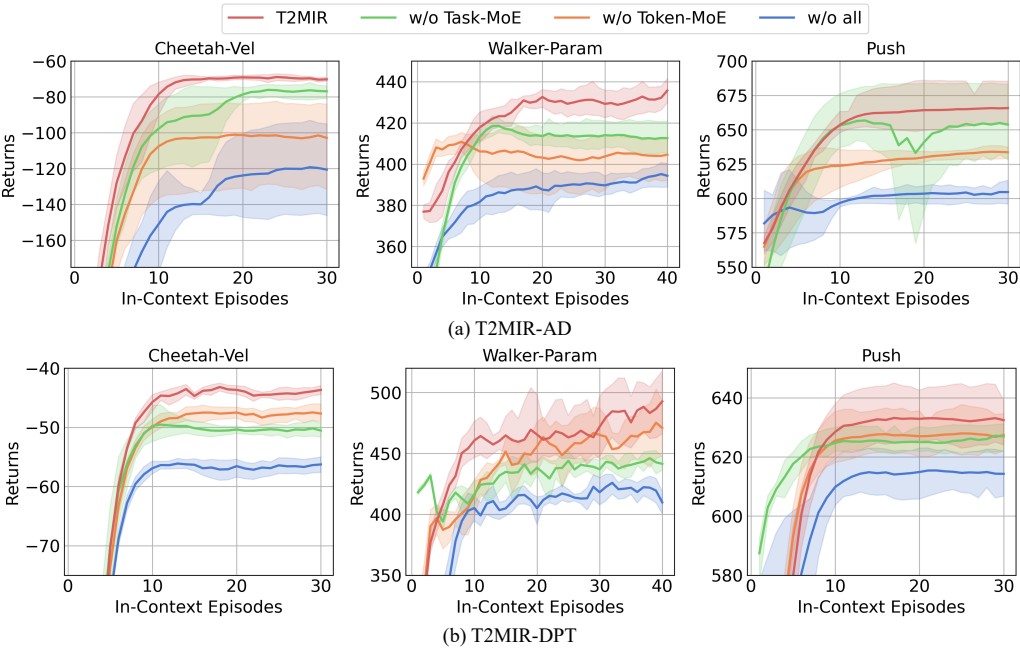

Figure 4: Ablation results of both T2MIR-AD and T2MIR-DPT architectures using Mixed datasets.

Table 2: Numerical results of ablation study on both T2MIR-AD and T2MIR-DPT using Mixed datasets, i.e., the best performance from Figure 4. Best result in **bold**.

| T2MIR-AD | **T2MIR** | w/o Task-MoE | w/o Token-MoE | AD |
|---|---|---|---|---|
| Cheetah-Vel | **-68.9**$_{\pm2.1}$ | $-76.1_{\pm3.7}$ | $-101.1_{\pm29.7}$ | $-119.2_{\pm30.4}$ |
| Walker-Param | **435.7**$_{\pm7.2}$ | $418.6_{\pm3.9}$ | $410.8_{\pm5.1}$ | $395.2_{\pm7.1}$ |
| Push | **665.8**$_{\pm13.9}$ | $656.8_{\pm17.8}$ | $634.0_{\pm4.0}$ | $604.7_{\pm6.3}$ |

| T2MIR-DPT | **T2MIR** | w/o Task-MoE | w/o Token-MoE | DPT |
|---|---|---|---|---|
| Cheetah-Vel | **-43.2**$_{\pm0.8}$ | $-49.5_{\pm3.4}$ | $-47.5_{\pm1.6}$ | $-56.1_{\pm1.0}$ |
| Walker-Param | **492.8**$_{\pm29.3}$ | $446.1_{\pm6.5}$ | $475.3_{\pm32.5}$ | $425.9_{\pm9.7}$ |
| Push | **633.2**$_{\pm8.6}$ | $627.2_{\pm4.0}$ | $628.0_{\pm7.2}$ | $615.4_{\pm6.0}$ |

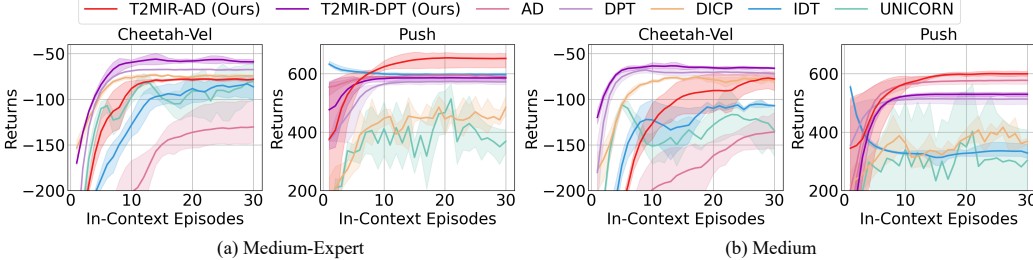

Figure 5: Test return curves of T2MIR against baselines using Medium-Expert and Medium datasets.

Table 3: Numerical results of T2MIR against baselines using Medium-Expert (top) and Medium (bottom) datasets, i.e., numerical results of best performance from Figure 5. Best result in **bold** and second best underline.

| Medium-Expert | **T2MIR-AD** | **T2MIR-DPT** | AD | DPT | DICP | IDT | UNICORN |
|---|---|---|---|---|---|---|---|
| Cheetah-Vel | $-77.8_{\pm1.5}$ | **-56.0**$_{\pm6.8}$ | $-130.3_{\pm24.2}$ | $\underline{-67.3}_{\pm0.6}$ | $-73.4_{\pm1.6}$ | $-83.1_{\pm14.4}$ | $-80.6_{\pm22.8}$ |
| Push | **654.4**$_{\pm24.3}$ | $586.3_{\pm9.5}$ | $595.4_{\pm3.5}$ | $573.6_{\pm8.5}$ | $485.5_{\pm29.2}$ | $\underline{632.9}_{\pm7.1}$ | $513.7_{\pm51.1}$ |

| Medium | **T2MIR-AD** | **T2MIR-DPT** | AD | DPT | DICP | IDT | UNICORN |
|---|---|---|---|---|---|---|---|
| Cheetah-Vel | $-76.6_{\pm11.4}$ | **-63.3**$_{\pm4.0}$ | $-135.8_{\pm19.2}$ | $\underline{-67.9}_{\pm3.2}$ | $-75.6_{\pm2.5}$ | $-105.3_{\pm4.1}$ | $-105.9_{\pm3.7}$ |
| Push | **600.5**$_{\pm6.8}$ | $530.0_{\pm6.7}$ | $\underline{577.7}_{\pm16.8}$ | $514.2_{\pm12.7}$ | $416.9_{\pm20.9}$ | $554.3_{\pm9.8}$ | $402.4_{\pm162.9}$ |

## 5.3 Robustness Study

**Robustness to Dataset Qualities.** We evaluate T2MIR on Medium-Expert and Medium datasets. As shown in Figure 5 and Table 3, both T2MIR-AD and T2MIR-DPT exhibit superior performance over baselines, validating their robustness when learning from varying qualities of datasets. Notably, most baselines suffer a significant performance drop when trained on Medium datasets, while T2MIR maintains satisfactory performance despite the lower data quality. It highlights T2MIR's appealing applicability in real-world scenarios where agents often learn from suboptimal data.

**Robustness to Expert Configurations.** We vary expert configurations in both MoE layers. For token-wise MoE that manages tokens of different modalities, we always activate one-third of the experts as there are three modalities and vary the total number of experts. Results in Figure 6-(a) and Figure 7-(a) show that a moderate configuration (2/6) yields the best performance. For task-wise MoE that manages task assignments, we always activate two experts while varying the total number of experts, following the common configuration in popular works. When testing the impact of expert configuration on one MoE structure, we keep the expert configuration of the other one unchanged. Results in Figure 6-(b) and Figure 7-(b) show that the performance slightly increases with more experts in total, and tends to saturate soon.

## 5.4 Visualization Insights into MoE Structure

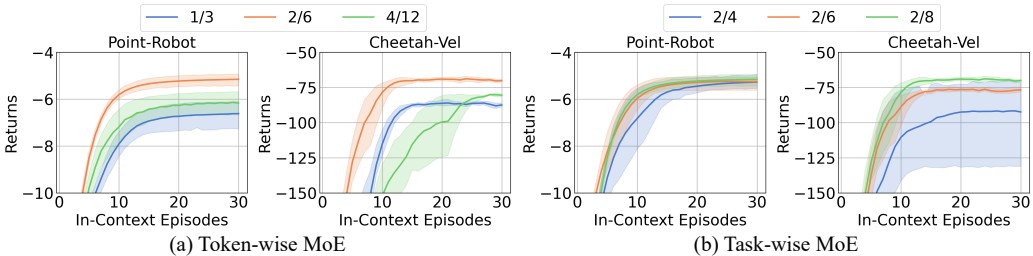

(a) Token-wise MoE         (b) Task-wise MoE

Figure 6: Analysis results of the number of selected experts against the total in token- and task-wise MoE on T2MIR-AD. For example, 2/6 denotes selecting the top 2 from 6 experts.

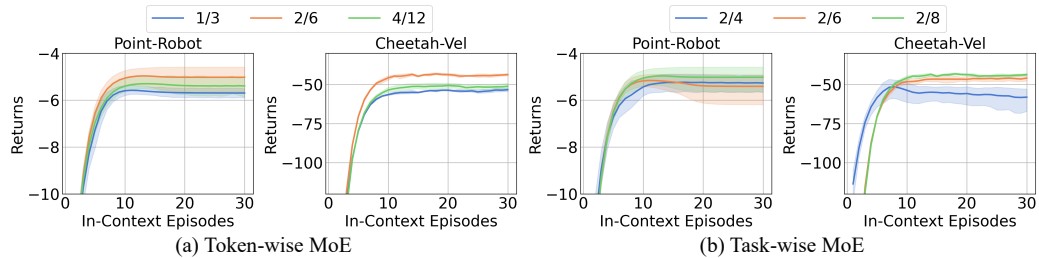

(a) Token-wise MoE         (b) Task-wise MoE

Figure 7: Analysis results of the number of selected experts against total experts in token- and task-wise MoE on T2MIR-DPT. For example, 2/6 denotes selecting the top 2 from 6 experts.

**Modality Clustering in Token-wise MoE.** In Figure 1 (left), the spatial proximity indicates similarity in expert assignments. It exhibits a clear clustering pattern that tokens from different modalities are routed to distinct experts, highlighting the successful utilization of the MoE structure to process tokens with distinct semantics.

**Task Clustering in Task-wise MoE.** In Figure 1 (right), points of similar color come from similar tasks. It forms a clear pattern in the router representation space, where similar tasks cluster closely and distinct tasks are widely separated. This confirms the effective use of the MoE structure to manage a broad task distribution. Appendix J gives more visualization on the two MoE layers.

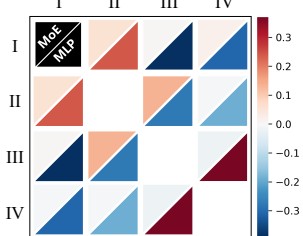

Figure 8: Cosine similarity of gradients between Point-Robot tasks in four quadrants (I-IV), comparing T2MIR-AD (MoE) with AD (MLP).

**Gradient Conflict Mitigation.** In Figure 8, AD (lower triangle) shows significant gradient conflict between opposing tasks (e.g., I vs. III, deep blue with a large negative cosine value) and fails to discriminate tasks (e.g., III vs. IV, deep red with a large positive cosine value). In contrast, T2MIR-AD (upper triangle) maintains nearly orthogonal gradients across diverse tasks, especially for opposing ones (e.g., I vs. III or II vs. IV, cosine value near 0). This comparison verifies T2MIR's advantage in both gradient conflict mitigation and task discrimination.

## 6   Conclusions, Limitations, and Future Work

In the paper, we introduce architectural advances of MoE to address input multi-modality and task diversity for ICRL. We propose a scalable framework, where a token-wise MoE processes distinct semantic inputs and a task-wise MoE handles a broad task distribution with contrastive routing. Improvements in generalization capacities highlight the potential impact of our method, and deep insights via visualization validate that our MoE design effectively addresses the associated challenges.

Though, our method is evaluated on widely adopted benchmarks in ICRL, with relatively lightweight datasets compared to popular large models. An urgent improvement is to evaluate on more complex environments such as XLand-MiniGrid [57, 10] with huge datasets, unlocking the scaling properties of MoE in ICRL domains. Another step is to deploy our method to vision-language-action (VLA) tasks [58, 59] that naturally involve more complex input multi-modality and task diversity.

## Acknowledgements

This work was supported in part by the National Natural Science Foundation of China (Nos. 62376122 and 72394363), and in part by the AI & AI for Science Project of Nanjing University.

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

# Appendix

# A  Algorithm Pseudocodes

Based on the implementations in Sec. 4, this section gives the brief procedures of T2MIR. Algorithm 1 and Algorithm 3 show the pipline of training T2MIR-AD and T2MIR-DPT, respectively. We train all components including token-wise MoE, task-wise MoE and their routers together with the main causal transformer network end to end. Then, Algorithm 2 and Algorithm 4 show the evaluation phase, where the agent can improve its performance on test tasks by interacting with the environments without any parameter updates.

---

**Algorithm 1:** Model Training of T2MIR-AD

---

**Input:** Training tasks $\mathcal{T}_{\text{train}}$ and corresponding offline datasets $\mathcal{D}_{\text{train}}$
        Causal transformer $\pi_\theta$;    Trajectory number $L$;    Prompt length $T$
        Batch size $B$;               Contrastive loss weight $w_{\text{con}}$

1  **for** *each iteration* **do**
2     Initialize query batch $\mathcal{B} = \{\}$, key batch $\mathcal{B}' = \{\}$
3     **for** $b = 1, 2, ..., B$ **do**
4         Sample a task $M_i \sim \mathcal{T}_{\text{train}}$ and obtain the corresponding dataset $\mathcal{D}_i$ from $\mathcal{D}_{\text{train}}$
5         Sample $L$ trajectories from $\mathcal{D}_i$
6         Sort $L$ trajectories by return and concatenate to $\tau_i = \{s_0, a_0, r_0, \ldots, s_T, a_T, r_T\}$
7         Sample another $L$ trajectories from $\mathcal{D}_i$ to obtain $\tau_i'$ as a positive key
8         Add $\tau_i$ and $\tau_i'$ to $\mathcal{B}$ and $\mathcal{B}'$ respectively
9     **end**
10    Get a query batch $\mathcal{B} = \{\tau\}_{b=1}^B$ and key batch $\mathcal{B}' = \{\tau'\}_{b=1}^B$
11    Autoregressively predict the actions with $\pi_\theta$ and compute loss in Eq. (10) using $\mathcal{B}$
12    Compute balance loss $\mathcal{L}_{\text{balance}}$ in Eq. (3)
13    Compute contrastive loss $\mathcal{L}_{\text{contrastive}}$ with key batch $\mathcal{B}'$ in Eq. (7)
14    Update $\pi_\theta$ by minimize loss $\mathcal{L} = \mathcal{L}(\theta) + \mathcal{L}_{\text{balance}} + w_{\text{con}} \cdot \mathcal{L}_{\text{contrastive}}$
15    Update momentum router in task-wise MoE in Eq. (8)
16 **end**

---

**Algorithm 2:** Model Evaluation of T2MIR-AD

---

**Input:** Test tasks $\mathcal{T}_{\text{test}}$;     Causal transformer $\pi_\theta$;    Evaluation episodes $N$
        History buffer $\mathcal{R}$;  History episode length $K$

1  **for** *each task $M_i \in \mathcal{T}_{test}$* **do**
2     Initialize history buffer $\mathcal{R} = \{\}$
3     **for** $n = 1, \cdots, N$ **do**
4         Reset env. and get feedback $s_0$
5         **for** *each timestep $t$* **do**
6             Predict action $a_t = \pi_\theta(s_t, \mathcal{R})$
7             Step env. and get feedback $s_{t+1}$ and $r_t$
8             Add $s_t, a_t, r_t$ to buffer $\mathcal{R}$
9         **end**
10        Sort $\mathcal{R}$ by episode return
11        **if** $n > K$ **then**
12            Drop the first episode in $\mathcal{R}$
13        **end**
14     **end**
15 **end**

---

---

**Algorithm 3:** Model Training of T2MIR-DPT

---

**Input:** Training tasks $\mathcal{T}_{\text{train}}$ and corresponding offline datasets $\mathcal{D}_{\text{train}}$
    Causal transformer $\pi_\theta$;   Prompt length $T$;   Expert policy $\pi^*$
    Batch size $B$;       Contrastive loss weight $w_{\text{con}}$

**1 for** *each task $M_i \in \mathcal{T}_{train}$* **do**
**2**   Obtain dataset $\mathcal{D}_i$ from $\mathcal{D}_{\text{train}}$
**3**   Get optimal action for states in $\mathcal{D}_i$ using expert policy $\pi_i^*$
**4**   Obtain query dataset $\mathcal{D}_i^q$
**5 end**
**6** Obtain query datasets $\mathcal{D}_{\text{train}}^q$
**7 for** *each iteration* **do**
**8**   Initialize query batch $\mathcal{B} = \{\}$, key batch $\mathcal{B}' = \{\}$
**9**   **for** $b = 1, 2, ..., B$ **do**
**10**    Sample a task $M_i \sim \mathcal{T}_{\text{train}}$
**11**    Obtain the corresponding dataset $\mathcal{D}_i$ from $\mathcal{D}_{\text{train}}$ and query dataset $\mathcal{D}_i^q$ from $\mathcal{D}_{\text{train}}^q$
**12**    Sample state-action pairs $(s, a^*)$ and $(s', a^{*\prime})$ from $\mathcal{D}_i^q$
**13**    Sample a trajectory from $\mathcal{D}_i$ and obtain $\tau_i = \{s_0, a_0, r_0, \ldots, s_T, a_T, r_T\}$
**14**    Sample another trajectory from $\mathcal{D}_i$ to obtain $\tau_i'$ as a positive key
**15**    Add $\big((s, a^*), \tau_i\big)$ and $\big((s', a^{*\prime}), \tau_i'\big)$ to $\mathcal{B}$ and $\mathcal{B}'$ respectively
**16**   **end**
**17**   Get a query batch $\mathcal{B} = \big\{\big((s, a^*), \tau\big)\big\}_{b=1}^B$ and key batch $\mathcal{B}' = \big\{\big((s', a^{*\prime}), \tau'\big)\big\}_{b=1}^B$
**18**   Autoregressively predict the optimal action with $\pi_\theta$ and compute loss in Eq. (11) using $\mathcal{B}$
**19**   Compute balance loss $\mathcal{L}_{\text{balance}}$ in Eq. (3)
**20**   Compute contrastive loss $\mathcal{L}_{\text{contrastive}}$ with key batch $\mathcal{B}'$ in Eq. (7)
**21**   Update $\pi_\theta$ by minimize loss $\mathcal{L} = \mathcal{L}(\theta) + \mathcal{L}_{\text{balance}} + w_{\text{con}} \cdot \mathcal{L}_{\text{contrastive}}$
**22**   Update momentum router in task-wise MoE in Eq. (8)
**23 end**

---

---

**Algorithm 4:** Model Evaluation of T2MIR-DPT

---

**Input:** Test tasks $\mathcal{T}_{\text{test}}$;   Causal transformer $\pi_\theta$;   Evaluation episodes $N$
    History buffer $\mathcal{R}$;   Episode history buffer $\tau$

**1 for** *each task $M_i \in \mathcal{T}_{test}$* **do**
**2**   Initialize episode history buffer $\tau = \{\}$
**3**   **for** $n = 1, \cdots, N$ **do**
**4**    Initialize history buffer $\mathcal{R} = \{\}$
**5**    Reset env. and get feedback $s_0$
**6**    **for** *each timestep $t$* **do**
**7**     Predict action $a_t = \pi_\theta(s_t, \tau)$
**8**     Step env. and get feedback $s_{t+1}$ and $r_t$
**9**     Add $s_t, a_t, r_t$ to buffer $\mathcal{R}$
**10**    **end**
**11**    $\tau \leftarrow \mathcal{R}$
**12**   **end**
**13 end**

---

# B  Regularization Loss for Balancing Expert Utilization

In this section, we provide detailed information about the regularization loss mentioned in Sec. 4.1. To prevent the gating network from converging to a state where it consistently assigns large weights to the same few experts, we adopt both importance loss and load-balancing loss as proposed by Shazeer et al. [25].

The importance loss aims to encourage all experts to have equal significance within the model, and the importance of an expert relative to a batch of training examples is defined as the sum of the gate values assigned to that expert:

$$\text{Imp}(h) = \sum_{x \in X} G_{\text{tok}}(x|h), \tag{12}$$

where $G_{\text{tok}}(x|h)$ represents the gate value for expert $h$ given input $x$. An additional loss term $\mathcal{L}_{\text{importance}}$ is then added to the overall loss function, which is calculated as the square of the coefficient of variation of the set of importance values multiplied by a hand-tuned scaling factor $w_{\text{imp}}$:

$$\mathcal{L}_{\text{importance}} = w_{\text{imp}} \cdot CV(\text{Imp}(h))^2. \tag{13}$$

Despite ensuring equal importance among experts, discrepancies may still arise in the number of input tokens each expert receives due to the top-$k$ activation mechanism. One expert might be assigned large weights for only a few tokens while another expert could receive small weights across many tokens but fail to activate. To address this issue, a load-balancing loss is introduced to ensure that each expert handles approximately the same number of input tokens.

To achieve this, a smooth estimator $\text{Load}(h)$ of the number of examples assigned to each expert. This estimator allows gradients to propagate through backpropagation, by utilizing the noise term in the gating function, which we omit in Eq. (2). Given a trainable weight of noise matrix $W_{noise}$, Eq. (2) is rewrite as

$$w_{\text{tok}}(i; h) = \text{softmax}(H(i|h))[i],$$
$$H(i|h) = \text{topk}\big(G_{\text{tok}}(i|h) + \text{StandardNormal}() \cdot \text{Softplus}(h \cdot W_{noise})_i\big). \tag{14}$$

Let $P(i; h)$ denote the probability that $w_{\text{tok}}(i; h)$ is non-zero, given a new random noise choice for element $i$, while keeping the already sampled noises for other elements constant. Note that, $w_{\text{tok}}(i; h)$ is non-zero if and only if $H(i|h)$ is greater than the $k^{th}$-greatest element of $H(\cdot|h)$ excluding itself. Thus, we have

$$P(i; h) = Pr\big(G_{\text{tok}}(i|h) + \text{StandardNormal}() \cdot \text{Softplus}(h \cdot W_{noise})_i > \text{kth\_excluding}(H(\cdot|h), k, i)\big), \tag{15}$$

where $\text{kth\_excluding}(H(\cdot|h), k, i)$ means the $k^{th}$-greatest element of $H(\cdot|h)$, excluding $i$-th element. Based on this, we can compute:

$$P(i; h) = \Phi\left(\frac{G_{\text{tok}}(i|h) - \text{kth\_excluding}(H(i|h), k, i)}{\text{Softplus}((h \cdot W_{noise})_i)}\right), \tag{16}$$

where $\Phi$ is the cumulative distribution function (CDF) of the standard normal distribution. Then, the estimated load for expert $i$ is given by:

$$\text{Load}(i; h) = \sum P(i; h). \tag{17}$$

Finally, the load-balancing loss is defined as the square of the coefficient of variation of the load vector, scaled by a hand-tuned parameter $w_{\text{load}}$:

$$\mathcal{L}_{\text{load}} = w_{\text{load}} \cdot CV(\text{Load}(h))^2. \tag{18}$$

Combining both losses yields the final regularization loss showed in Eq. (3) used in our model which helps maintain balanced utilization of experts during training, preventing any single expert from dominating the computation.

# C Contrastive Learning for Task-wise MoE Router

In this section, we give the proof of Theorem 1 based on a lemma as follows.

**Lemma 1.** *Give a task from the task distribution $M \sim P(M)$, let $\bar{h} = f(\tau)$ as the average of hidden state after self-attention calculation of task $M$, $z \sim G_{task}(z|\bar{h})$. Then, we have*

$$\frac{p(M|z)}{p(M)} = \mathbb{E}_{\bar{h}}\left[\frac{p(z|\bar{h})}{p(z)}\right]. \tag{19}$$

*Proof.*

$$\begin{aligned}
\frac{p(M|z)}{p(M)} &= \frac{p(z|M)}{p(z)} \\
&= \int_{\bar{h}} \frac{p(z|\bar{h})p(\bar{h}|M)}{p(z)} \, d\bar{h} \\
&= \mathbb{E}_{\bar{h}}\left[\frac{p(z|\bar{h})}{p(z)}\right].
\end{aligned} \tag{20}$$

The proof is completed. □

**Theorem 1.** *Let $\mathcal{M}$ denote a set of tasks following the task distribution $P(M)$, and $|\mathcal{M}| = N$. $M \in \mathcal{M}$ is a given task. Let $\bar{h} = f(\tau)$, $z \sim G_{task}(\cdot|\bar{h})$, and $e(\bar{h}, z) = \frac{p(z|\bar{h})}{p(z)}$, where $\tau$ is a trajectory from task $M$ and $\bar{h}$ is average hidden state of all tokens after the self-attention calculation function $f(\cdot)$. Let $\bar{h}'$ denote the average hidden state generated by any task $M' \in \mathcal{M}$, then we have*

$$I(z; M) \geq \log N + \mathbb{E}_{\mathcal{M},z,\bar{h}}\left[\log \frac{e(\bar{h}, z)}{\sum_{M' \in \mathcal{M}} e(\bar{h}', z)}\right]. \tag{21}$$

*Proof.* Using Lemma 1 and Jensen's inequality, we have

$$\begin{aligned}
\mathbb{E}_{\mathcal{M},z,\bar{h}}\left[\log \frac{e(\bar{h}, z)}{\sum_{M' \in \mathcal{M}} e(\bar{h}', z)}\right] &= \mathbb{E}_{\mathcal{M},z,\bar{h}}\left[\log \frac{\frac{p(z|\bar{h})}{p(z)}}{\frac{p(z|\bar{h})}{p(z)} + \sum_{M' \in \mathcal{M}\backslash M} \frac{p(z|\bar{h}')}{p(z)}}\right] \\
&= \mathbb{E}_{\mathcal{M},z,\bar{h}}\left[-\log\left(1 + \frac{p(z)}{p(z|\bar{h})} \sum_{M' \in \mathcal{M}\backslash M} \frac{p(z|\bar{h}')}{p(z)}\right)\right] \\
&\approx \mathbb{E}_{\mathcal{M},z,\bar{h}}\left[-\log\left(1 + \frac{p(z)}{p(z|\bar{h})} (N-1)\mathbb{E}_{M' \in \mathcal{M}\backslash M}\left[\frac{p(z|\bar{h}')}{p(z)}\right]\right)\right] \\
&= \mathbb{E}_{\mathcal{M},z,\bar{h}}\left[-\log\left(1 + \frac{p(z)}{p(z|\bar{h})} (N-1)\right)\right] \\
&= \mathbb{E}_{\mathcal{M},z,\bar{h}}\left[\log\left(\frac{1}{1 + \frac{p(z)}{p(z|\bar{h})}(N-1)}\right)\right] \\
&\leq \mathbb{E}_{\mathcal{M},z,\bar{h}}\left[\log\left(\frac{1}{\frac{p(z)}{p(z|\bar{h})}N}\right)\right] \\
&\leq \mathbb{E}_{\mathcal{M},z}\left[\log \mathbb{E}_{\bar{h}}\left[\frac{p(z|\bar{h})}{p(z)}\right]\right] - \log N \\
&= \mathbb{E}_{\mathcal{M},z}\left[\log \frac{p(M|z)}{p(M)}\right] - \log N \\
&= I(z; M) - \log N.
\end{aligned} \tag{22}$$

Thus, we complete the proof. □

# D  The Details of Environments and Dataset Construction

In this section, we show details of the evaluation environments over a variety of benchmarks, as well as the collection of offline datasets on these environments.

## D.1  The Details of Environments

We evaluate T2MIR and all the baselines on classical benchmarks including discrete environment commonly used in ICRL [12], the 2D navigation, the multi-task MuJoCo control [60] and the Meta-World [61]. More specifically, we evaluate all tested methods on the following environments as

- **DarkRoom**: a discrete environment where the agent navigates in a $10 \times 10$ grid to find the goal. The observation is the 2D coordinate of the agent, and the actions are left, right, up, down, and stay. The agent is started from $[0, 0]$ to find the goal which is uniformly sampled from the grid. The reward is sparse and only $r = 1$ when the agent is at the goal, and $r = 0$ otherwise. It provides 100 tasks, from which we randomly sample 80 tasks as training tasks and hold out the remaining 20 for evaluation. The maximal episode step is set to 100.

- **Point-Robot**: a problem of a point agent navigating to a given goal position in the 2D space. The observation is the 2D coordinate of the robot. The action space is $[-0.1, 0.1]^2$ with each dimension corresponding to the moving distance in the horizontal and vertical directions. The reward function is defined as the negative distance between the point agent and the goal location. Tasks differ in goal positions that are uniformly distributed in a unit square, resulting in the variation of the reward functions. We randomly sample 45 goals as training tasks, and another 5 goals for evaluation. The start position is sampled uniformly from $[-0.1, 0.1]^2$ for each learning episode and the maximal episode step is set to 20.

- **Cheetah-Vel**: a multi-task MuJoCo continuous control environment in which the reward function differs across tasks. It requires a planar cheetah robot to run at a particular velocity in the positive $x$-direction. The observation space is $\mathbb{R}^{20}$, which comprises the position and velocity of the cheetah, the angle and angular velocity of each joint. The action space is $[-1, 1]^6$ with each dimension corresponding to the torque of each joint. The reward function is negatively correlated with the absolute value between the current velocity of the agent and the goal, plus the control cost. The goal velocities are uniformly sampled from the distribution $U[0.075, 3.0]$. We randomly sample 45 goals as training tasks, and another 5 goals for evaluation. The maximal episode step is set to 200.

- **Ant-Dir**: a multi-task MuJoCo continuous control environment in which the reward function differs across tasks. It requires a quadruped ant robot to run in a target direction. The observation space is $\mathbb{R}^{27}$, which comprises the position and velocity of the ant robot, together with the angle and angular velocity of 8 joints. The action space is $[-1, 1]^8$ with each dimension corresponding to the torque of each joint. The reward function is positively correlated with the velocity of the ant robot in the target direction, plus the control cost. The target directions are uniformly sampled from $U[0, 2\pi]$. We randomly sample 45 target directions as training tasks, and another 5 for evaluation. The maximal episode step is set to 200.

- **Walker-Param**: a multi-task MuJoCo benchmark where tasks differ in state transition dynamics. It need to control a two-legged walker robot to run as fast as possible with varying environment dynamics. The observation space is $\mathbb{R}^{17}$ and the action space is $[-1, 1]^6$. The reward function is proportional to the running velocity in the positive $x$-direction, which remains consistent for different tasks. The physical parameters of body mass, inertia, damping, and friction are randomized across tasks. We randomly sample 45 physical parameters as training tasks, and another 5 for evaluation. The maximal episode step is set to 200.

- **Reach**, **Push**: two typical environments from the robotic manipulation benchmark Meta-World. Reach and Push control a robotic arm to reach a goal location in 3D space and to push the puck to a goal, respectively. The observation space is $\mathbb{R}^{39}$, which contains current state, previous state, and the goal. We only use the current state vector, thus the observation space is modified to $\mathbb{R}^{18}$ without goal information. The action space is $[-1, 1]^4$. Tasks differ in goal positions which are uniformly sampled, resulting in the variation of reward functions. The initial position of object is fixed across all tasks. We randomly sample 45 goals as training tasks, and another 5 goals for evaluation. The maximal episode step is set to 100.

- **ML10**: contains 10 different robotics manipulation tasks for training and another 5 different tasks for evaluation. Specifically, we randomly sample 5 goals for each training task and 1 goal for

Table 4: Hyperparameters of SAC used to collect multi-task datasets.

| Environments | Training steps | Warmup steps | Save frequency | Learning rate | Soft update | Discount factor | Entropy ratio |
|---|---|---|---|---|---|---|---|
| Point-Robot | 2000 | 100 | 20 | 3e-4 | 0.005 | 0.99 | 0.2 |
| Cheetah-Vel | 250000 | 2000 | 10000 | 3e-4 | 0.005 | 0.99 | 0.2 |
| Cheetah-Vel-3_Cluster | 400000 | 2000 | 10000 | 3e-4 | 0.005 | 0.99 | 0.2 |
| Walker-Param | 1000000 | 2000 | 10000 | 3e-4 | 0.005 | 0.99 | 0.2 |

Table 5: Hyperparameters of PPO used to collect multi-task datasets.

| Environments | total_timesteps | n_steps | learning_rate | batch_size | n_epochs | $\gamma$ |
|---|---|---|---|---|---|---|
| Reach | 400000 | 2048 | 3e-4 | 64 | 10 | 0.99 |
| Push | 1000000 | 2048 | 3e-4 | 64 | 10 | 0.99 |

each evaluation task, resulting in 50 goals for training and 5 goals for evaluation. We use the same setting for each robotic manipulation task as in Reach and Push. The maximal episode step is set to 100.

Furthermore, we give the details of the heterogeneous version `Cheetah-Vel` we used in Appendix J.

- **Cheetah-Vel-3_Cluster**: a heterogeneous version of `Cheetah-Vel`, where the goal velocities are different. We sample the goal velocities from three Gaussian distributions: $\mathcal{N}(0.5, 0.15^2)$, $\mathcal{N}(1.5, 0.15^2)$, $\mathcal{N}(2.5, 0.15^2)$. From each Gaussian distribution, we sample 14 tasks for training and 2 tasks for evaluation, resulting in 42 training tasks and 6 test tasks.

Note that we construct different tasks by setting different parameters for the environment, we cannot access these parameters (e.g., goal velocities in `Cheetah-Vel`) during training following common in-context RL settings.

## D.2 The Details of Datasets

For discrete environment DarkRoom, we use the expert policy to collect datasets by progressively reducing the noise, as in [14]. For Point-Robot and MuJoCo environments, we employ the soft actor-critic (SAC)[62] algorithm to train a policy independently for each task, and the detailed hyperparameters are shown in Table 4. For Meta-World environments, we use the Proximal Policy Optimization (PPO)[63] algorithm implementation provided by Stable Baselines 3[64], and the detailed hyperparameters are shown in Table 5. During training, we periodically save the policy checkpoints and use them to generate various qualities of offline datasets as

- **Mixed**: using all policy checkpoints to generate datasets. We use each checkpoint to generate same number of episodes, e.g., using one checkpoint to generate one episode.
- **Medium-Expert**: using the policy checkpoints whose performance is below 80% of the final achieved level to generate datasets. The max performance in Medium-Expert datasets is about 80% of that in Mixed datasets.
- **Medium**: using the policy checkpoints whose performance is below 50% of the final achieved level to generate datasets. The max performance in Medium datasets is about 50% of that in Mixed datasets.

For query state-action pairs used in DPT and T2MIR-DPT, we use the expert policies that achieve 100%, 80% and 50% performance to provide actions for states in Mixed, Medium-Expert and Medium datasets, respectively. As we find the actions provided by expert policies contain too many values outside the boundary of the action space in Meta-World environments, we just use the state-action pairs in offline datasets as query datasets for Reach and Push.

# E    Baseline Methods

This section gives the details of the five representative baselines, including four ICRL approaches and one context-based offline meta-RL method. These baselines are thoughtfully selected as they are representative in distinctive categories. Furthermore, since our proposed T2MIR method belongs to the ICRL category, we incorporate more methods from this class as baselines for a comprehensive comparison. The detailed descriptions of these baselines are as follows:

- **AD** [11], is the first method to achieve ICRL through sequential modeling of offline historical data using an imitation loss. By employing a causal transformer with sufficiently large across-episodic contexts to imitate gradient-based RL algorithms, AD learns improved policies for new tasks entirely in context without requiring external parameters updating.

- **DPT** [12], is an ICRL method that models contextual trajectories via supervised learning to predict optimal actions. After pretraining on offline datasets, DPT provides contextual learning capabilities for new tasks, enabling online exploration and offline decision-making given query states and contextual prompts.

- **IDT** [13], is an ICRL method that simulates high-level trial-and-error processes. It introduces an innovative architecture comprising three modules: Making Decisions, Decisions to Go, and Reviewing Decisions. These modules generate high-level decisions in an autoregressive manner to guide low-level action selection. Built upon transformer models, IDT can address complex decision-making in long-horizon tasks while reducing computational costs.

- **DICP** [15], combines model-based reinforcement learning with previous ICRL algorithms like AD and IDT. By jointly learning environment dynamics and policy improvements in context, DICP employs a transformer architecture to perform model-based planning and decision-making without updating model parameters Furthermore, DICP generates actions by simulating potential future states and predicting long-term returns.

- **UNICORN** [42], is a context-based offline meta-RL algorithm grounded in information theory. It proposes a unified information-theoretic framework, which focuses on optimizing different approximation bounds of the mutual information objective between task variables and their latent representations. Leveraging the information bottleneck principle, it derives a novel general and unified task representation learning objective, enabling the acquisition of robust task representations.

To ensure a fair comparison, all baselines are adjusted to an aligned setting, where test datasets are not available for policy evaluation. We also standardize the size and quality of the offline datasets for all baselines.

In our experimental setup, AD faces difficulties in distilling effective policy improvement operators due to the scarcity of available offline historical data. Thus, we use an enhanced version of AD that incorporates a trajectory-ranking mechanism based on cumulative rewards, inspired by AT [65].

# F    Network Architecture and Hyperparameters of T2MIR

This section gives details of the architecture and hyperparameters of T2MIR implementations as follows.

**Router Network.** We implement the router network using a multi-layer perceptron (MLP) architecture without bias. Specifically, the router contains two linear layers with Tanh as the activation function between them. The first linear maps hidden state $h$ to a $n\_experts$-dim vector along with the Tanh activation, and the second linear maps the $n\_experts$-dim vector to another $n\_experts$-dim vector as the output of router.

**Architecture of T2MIR.** We implement the T2MIR-AD based on causal transformer akin to AMAGO [45] with Flash Attention [56]. For T2MIR-DPT, we build our T2MIR framework upon the transformer architecture in DPT [12], flatten the trajectories into state-action-reward sequence as input. Specifically, we employ separate embeddings for states, action, and rewards, adding a learnable positional embedding based on timesteps. Each block employs a multi-head self-attention module followed by a feedforward network or a MoE layer with GELU activation [66]. For action prediction, we employ a linear to map the output of transformer blocks to an action vector with a Tanh activation.

Table 6: Hyperparameters in training process of T2MIR-AD using Mixed datasets.

| Hyperparameters | DarkRoom | Point-Robot | Cheetah-Vel | Walker-Param | Reach | Push |
|---|---|---|---|---|---|---|
| training steps | 300000 | 100000 | 100000 | 100000 | 100000 | 100000 |
| learning rate | 3e-4 | 3e-4 | 3e-4 | 3e-4 | 5e-5 | 3e-4 |
| Prompt length | 400 | 80 | 800 | 800 | 400 | 400 |
| token experts $K_1$ | 6 | 6 | 6 | 6 | 6 | 6 |
| task experts $K_2$ | 12 | 8 | 8 | 8 | 8 | 8 |
| InfoNCE weight | 0.01 | 0.01 | 0.01 | 0.01 | 0.01 | 0.01 |
| MoE layer position | top | top | top | top | top | top |
| Momentum ratio $\beta$ | 0.995 | 0.995 | 0.995 | 0.995 | 0.995 | 0.995 |

Table 7: Hyperparameters in training process of T2MIR-DPT using Mixed datasets.

| Hyperparameters | DarkRoom | Point-Robot | Cheetah-Vel | Walker-Param | Reach | Push |
|---|---|---|---|---|---|---|
| training steps | 300000 | 100000 | 100000 | 100000 | 100000 | 100000 |
| learning rate | 3e-4 | 3e-4 | 3e-4 | 3e-4 | 5e-5 | 3e-4 |
| Prompt length | 100 | 20 | 200 | 200 | 100 | 100 |
| token experts $K_1$ | 6 | 6 | 6 | 6 | 6 | 6 |
| task experts $K_2$ | 8 | 8 | 8 | 8 | 8 | 8 |
| InfoNCE weight | 0.001 | 0.001 | 0.01 | 0.01 | 0.01 | 0.001 |
| MoE layer position | top | top | top | top | top | top |
| Momentum ratio $\beta$ | 0.995 | 0.995 | 0.995 | 0.995 | 0.995 | 0.995 |

Table 6 and Table 7 show the detailed hyperparameters used for T2MIR-AD and T2MIR-DPT using Mixed datasets, respectively.

**Compute.** We train our models on one Nvidia RTX4080 GPU with the Intel Core i9-10900X CPU and 256G RAM. The training process takes about 0.5-3 hours, depending on the complexity of the environments. Compared with T2MIR-free backbones, T2MIR takes a lightly higher computational cost during the training process, but it requires less computational resources during task inference.

# G  Analysis of Hyperparameters and T2MIR Architecture

## G.1  Hyperparameter Analysis

**Analysis of Expert Configurations on T2MIR-DPT.** We also investigate different expert configurations on T2MIR-DPT, as shown in Figure 7. Table 8 and Table 9 give the numerical results of T2MIR with different expert configurations in token- and task-wise MoE, respectively. For token-wise MoE, as same as T2MIR-AD, the results show that moderate configuration (2/6) outperforms both smaller (1/3) and larger (4/12) settings. For task-wise MoE, the performance improves with the increase number of total experts, and tends to be stable.

**Weight of InfoNCE Loss.** We investigate the influence of the loss weight of InfoNCE loss in task-wise MoE, the results are shown in Figure 9 and Table 10. The performance of T2MIR is not sensitive to small weights. But when the InfoNCE loss weight is set to $0.1$, it gains an obvious decrease. We suppose this is because the final loss of $\mathcal{L}_{\text{contrastive}}$ in Eq. (7) has same magnitude with imitation loss $\mathcal{L}(\theta)$ in Eq. (10) and Eq. (11), and it disturbs the overall learning process.

## G.2  Architecture Analysis

**Position of MoE Layer.** Further more, we explore how the placement of the MoE layer affects performance. We substitute the feedforward network in one transformer block with MoE layer and

Table 8: Numerical results of T2MIR with varying expert numbers in token-wise MoE. Best in **bold**.

| T2MIR-AD | 1/3 | 2/6 | 4/12 | T2MIR-DPT | 1/3 | 2/6 | 4/12 |
|---|---|---|---|---|---|---|---|
| Point-Robot | $-6.6_{\pm0.7}$ | $\mathbf{-5.2}_{\pm0.3}$ | $-6.1_{\pm0.5}$ | Point-Robot | $-5.6_{\pm0.2}$ | $\mathbf{-5.0}_{\pm0.5}$ | $-5.3_{\pm0.3}$ |
| Cheetah-Vel | $-86.1_{\pm2.2}$ | $\mathbf{-68.9}_{\pm2.1}$ | $-80.1_{\pm0.5}$ | Cheetah-Vel | $-53.3_{\pm1.3}$ | $\mathbf{-43.2}_{\pm0.8}$ | $-50.7_{\pm1.9}$ |

Table 9: Numerical results of T2MIR with varying expert numbers in task-wise MoE. Best in **bold**.

| T2MIR-AD | 2/4 | 2/6 | 2/8 | T2MIR-DPT | 2/4 | 2/6 | 2/8 |
|---|---|---|---|---|---|---|---|
| Point-Robot | $-5.3_{\pm0.3}$ | $-5.2_{\pm0.3}$ | $\mathbf{-5.2}_{\pm0.3}$ | Point-Robot | $-5.2_{\pm0.4}$ | $-5.2_{\pm0.2}$ | $\mathbf{-5.0}_{\pm0.5}$ |
| Cheetah-Vel | $-91.7_{\pm27.9}$ | $-76.3_{\pm1.7}$ | $\mathbf{-68.9}_{\pm2.1}$ | Cheetah-Vel | $-51.5_{\pm7.4}$ | $-45.9_{\pm2.1}$ | $\mathbf{-43.2}_{\pm0.8}$ |

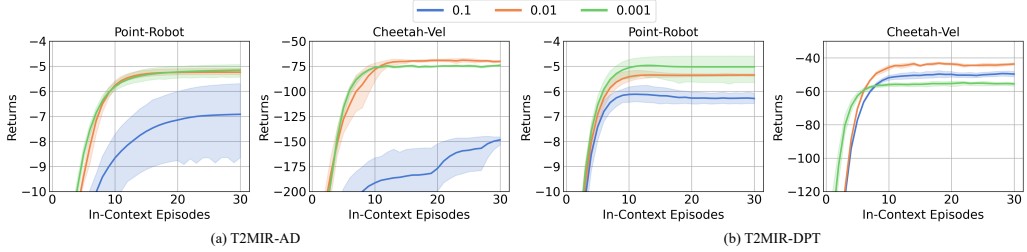

Figure 9: Test return curves of T2MIR with different contrastive loss weights.

Table 10: Numerical results of T2MIR with different loss weights of InfoNCE loss, i.e., numerical results of best performance from Figure 9. Best result in **bold**.

| T2MIR-AD | 0.1 | 0.01 | 0.001 | T2MIR-DPT | 0.1 | 0.01 | 0.001 |
|---|---|---|---|---|---|---|---|
| Point-Robot | $-6.9_{\pm1.6}$ | $\mathbf{-5.2}_{\pm0.1}$ | $-5.2_{\pm0.3}$ | Point-Robot | $-6.1_{\pm0.4}$ | $-5.4_{\pm0.1}$ | $\mathbf{-5.0}_{\pm0.5}$ |
| Cheetah-Vel | $-148.4_{\pm5.0}$ | $\mathbf{-68.9}_{\pm2.1}$ | $-74.1_{\pm1.0}$ | Cheetah-Vel | $-49.4_{\pm1.6}$ | $\mathbf{-43.2}_{\pm0.8}$ | $-54.9_{\pm1.3}$ |

study three settings of different positions: `bottom`, `middle` and `top`, where the MoE layer is used in the first, $\frac{L}{2}$-th and last transformer block, respectively. Figure 10 and Table 11 present the results, where substituting the feedforward network in the last transformer block with MoE layer (`top`) gets more stable performance.

**Importance of Balance and InfoNCE Loss.** We conduct further ablation studies to investigate the respective impacts of additional losses used to balance the experts. We compare T2MIR-AD to three ablations: 1) `w/o balance_loss`, omitting balance loss (Eq. (3)) in token-wise MoE, while retaining InfoNCE loss in task-wise MoE; 2) `w/o infonce_loss`, omitting InfoNCE loss (Eq. (7)) in task-wise MoE, while retaining balance loss in token-wise MoE; and 3) `w/o aux_loss`, omitting both balance loss and InfoNCE loss. For the ablation studies, we just omit the additional losses while keeping the token- and task-wise MoE architectures. The numerical results in Table 12 show that both balance loss and InfoNCE loss play significant roles in T2MIR's superior performance. Ablating the balance loss will cause a bit more performance degradation than ablating the InfoNCE loss.

**Architecture of Token-wise MoE.** To quantify the challenge incurred by the semantic discrepancy among states, actions, and rewards, we completely isolate the multi-modality issue and manually design a MoE structure where state-action-reward tokens are routed to three experts by a heuristic gating scheme, i.e., one expert for one modality. The results in Table 13 show that the heuristic gating mechanism contributes to performance improvement, while T2MIR achieves a more pronounced enhancement by automatically routing state-action-reward tokens. The enhancement achieved by transitioning from heuristic routing to automatic routing may stem from a more flexible approach to processing the integrated information generated after the attention mechanism. The results again demonstrate the significant challenge induced by the semantic discrepancy across multi-modal tokens in the state-action-reward sequence, and the significant superiority of token-wise MoE architecture.

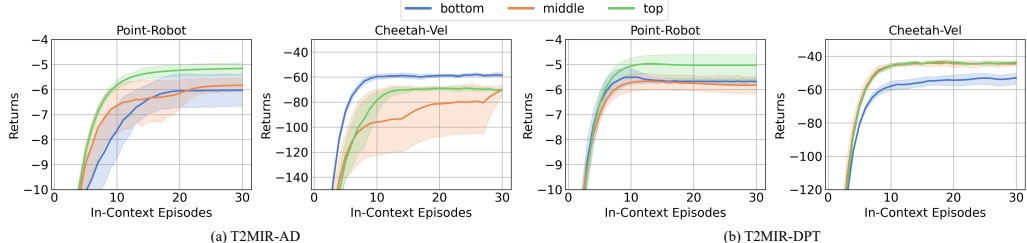

Figure 10: Test return curves of T2MIR with different positions of MoE layer.

Table 11: Numerical results of T2MIR with different positions of MoE layer, i.e., numerical results of best performance from Figure 10. Best result in **bold**.

| T2MIR-AD | bottom | middle | top | T2MIR-DPT | bottom | middle | top |
|---|---|---|---|---|---|---|---|
| Point-Robot | $-6.0_{\pm 0.8}$ | $-5.8_{\pm 0.3}$ | **-5.2**$_{\pm 0.3}$ | Point-Robot | $-5.5_{\pm 0.3}$ | $-5.7_{\pm 0.4}$ | **-5.0**$_{\pm 0.5}$ |
| Cheetah-Vel | **-57.9**$_{\pm 1.6}$ | $-70.2_{\pm 5.4}$ | $-68.9_{\pm 2.1}$ | Cheetah-Vel | $-52.9_{\pm 4.0}$ | $-43.8_{\pm 2.2}$ | **-43.2**$_{\pm 0.8}$ |

Table 12: Numerical results of ablation study on the importance of balance and InfoNCE loss on T2MIR-AD using Mixed datasets. Best result in **bold**.

| Environments | **T2MIR-AD** | w/o infonce_loss | w/o balance_loss | w/o aux_loss |
|---|---|---|---|---|
| Cheetah-Vel | **-68.9**$_{\pm 2.1}$ | $-76.7_{\pm 3.5}$ | $-92.8_{\pm 1.6}$ | $-112.4_{\pm 30.4}$ |
| Walker-Param | **435.7**$_{\pm 7.2}$ | $417.0_{\pm 2.6}$ | $416.0_{\pm 5.3}$ | $393.2_{\pm 6.7}$ |

Table 13: Numerical results of different architectures of token-wise MoE on T2MIR-AD using Mixed datasets. Best result in **bold**.

| Environments | **T2MIR-AD** | heuristic routing | AD |
|---|---|---|---|
| Cheetah-Vel | **-68.9**$_{\pm 2.1}$ | $-91.1_{\pm 30.4}$ | $-119.2_{\pm 30.4}$ |
| Walker-Param | **435.7**$_{\pm 7.2}$ | $407.1_{\pm 10.5}$ | $395.2_{\pm 7.1}$ |

## H  Robustness to OOD Setups

### H.1  Robustness to OOD test tasks

We conduct new experiments to evaluate T2MIR's capability to handle OOD test tasks on Cheetah-Vel against baselines. The goal velocities of training tasks are uniformly sampled from $U[0.075, 3]$, but during testing we sample OOD tasks from $U[3.1, 3.3]$. The results in Table 14 demonstrate T2MIR's consistent superiority even when dealing with OOD test tasks.

### H.2  Robustness to OOD Offline Prompt Data

In our main experiments in Sec. 5.1, we evaluate T2MIR and baselines via direct online interaction with the test environment, without access to any offline data. That is, the prompts or contexts are generated from online environment interactions to infer task beliefs during testing. In this section, we have conducted new experiments to evaluate T2MIR-DPT's robustness to OOD offline data as the test prompt. Specifically, we employ a random behavior policy to interact with the environment and collect one trajectory as the offline prompt. The results in Table 15 demonstrate T2MIR's consistent superiority even when prompted with OOD offline data.

## I  More Experiments on Ant-Dir and Meta-World

In addition, we have conducted new evaluations on the challenging Ant-Dir in MuJoCo and ML10 in Meta-World to demonstrate T2MIR's ability to these harder suites. Details about the two environments

Table 14: Numerical results of T2MIR against baselines on OOD test tasks using Mixed datasets. Best result in **bold** and second best underline.

| Environment | **T2MIR-AD** | **T2MIR-DPT** | AD | DPT | DICP | IDT | UNICORN |
|---|---|---|---|---|---|---|---|
| Cheetah-Vel | $-117.3_{\pm 4.4}$ | **$-71.0_{\pm 0.7}$** | $-158.8_{\pm 47.1}$ | $-78.5_{\pm 2.6}$ | $-109.1_{\pm 2.1}$ | $-114.3_{\pm 2.3}$ | $-88.6_{\pm 2.1}$ |

Table 15: Numerical results of T2MIR-DPT against DPT with OOD offline prompt data using Mixed datasets.

| Environments | T2MIR-DPT | DPT |
|---|---|---|
| Point-Robot | $-5.0_{\pm 0.2}$ | $-5.9_{\pm 0.1}$ |
| Cheetah-Vel | $-35.0_{\pm 2.9}$ | $63.2_{\pm 4.4}$ |
| Walker-Param | $424.1_{\pm 17.3}$ | $419.0_{\pm 11.5}$ |

Table 16: Numerical results of T2MIR on Ant-Dir in MuJoCo and ML10 in Meta-World. Best result in **bold** and second best underline.

| Environments | **T2MIR-AD** | **T2MIR-DPT** | AD | DPT | DICP | IDT | UNICORN |
|---|---|---|---|---|---|---|---|
| Ant-Dir | $681.5_{\pm 30.4}$ | **$708.6_{\pm 24.5}$** | $432.2_{\pm 42.5}$ | $634.1_{\pm 46.7}$ | $690.3_{\pm 20.4}$ | $643.7_{\pm 34.5}$ | $450.9_{\pm 10.2}$ |
| ML10 | **$318.1_{\pm 11.3}$** | $291.3_{\pm 21.9}$ | $277.0_{\pm 39.0}$ | $230.6_{\pm 25.1}$ | $215.9_{\pm 11.2}$ | $223.3_{\pm 13.3}$ | $193.6_{\pm 10.7}$ |

can be found in Appendix D. The numerical results in Table 16 show the promising scalability and the consistent superiority of T2MIR over various baselines.

## J   More Visualization Insights

**t-SNE Visualization on Push.** We also visualize the multi-modal property and task clustering in T2MIR-AD on Push, as shown in Figure 12. The visualization results consistently exhibit the ability of token-wise MoE to process token from different modalities with different experts, and task-wise MoE to distribute trajectories from different tasks to different experts.

**Heterogeneous Setting.** For a further study of the task clustering ability of T2MIR, we construct a heterogeneous version `Cheetah-Vel` inspired by MᵢLEₜ [67]. We split the velocities of tasks into three intervals, constructing `Cheetah-Vel-3_Cluster` environment. Figure 11 presents the test return curves of T2MIR and baselines using Mixed datasets. We demonstrate the probability of task assignments to some experts, as shown in Figure 13,

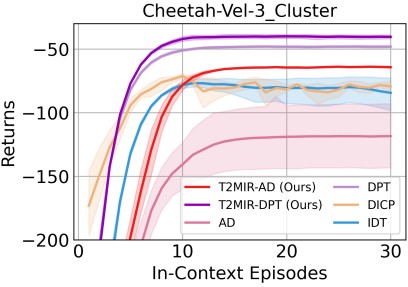

Figure 11: Test return curves of T2MIR against baselines on Cheetah-Vel-3_Clustering using Mixed datasets.

the velocities from three different distributions are divided by dashed lines. The expert-1 dominates at lower speeds, the expert-2 dominates at higher speeds, and it exhibits a mixture of different experts for medium speeds, with expert-1 being dominant. Moreover, there are switched among different experts at the boundaries between different velocity distributions. The results indicate that distinct experts dominate different tasks sampled from the three distributions, which further validates our motivation to employ token-wise MoE.

## K   Empirical Evidence of Gradient Conflicts

In addition to the results in Figure 8, we also provide quantification of the gradient conflict issue. The results in Table 17 present the proportion of task pairs with negative correlations (cosine similarity $< -0.05$) among all pairs of tasks, which demonstrate T2MIR's significant superiority in mitigating gradient conflicts.

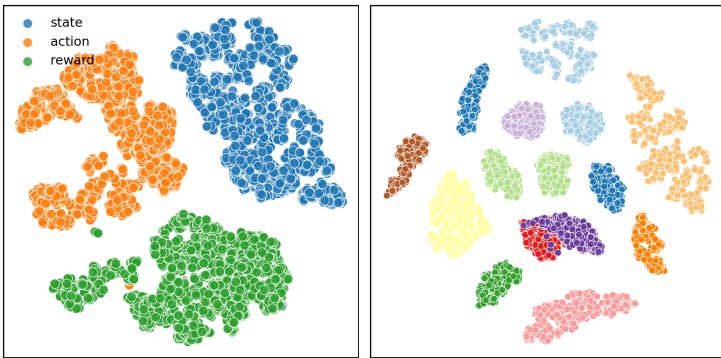

Figure 12: t-SNE visualization of expert assignments on Push. Left: token-wise MoE enables different experts to process three-modality tokens. Right: task-wise MoE effectively manages a task distribution, where trajectories from same task are prone to be distributed to same experts.

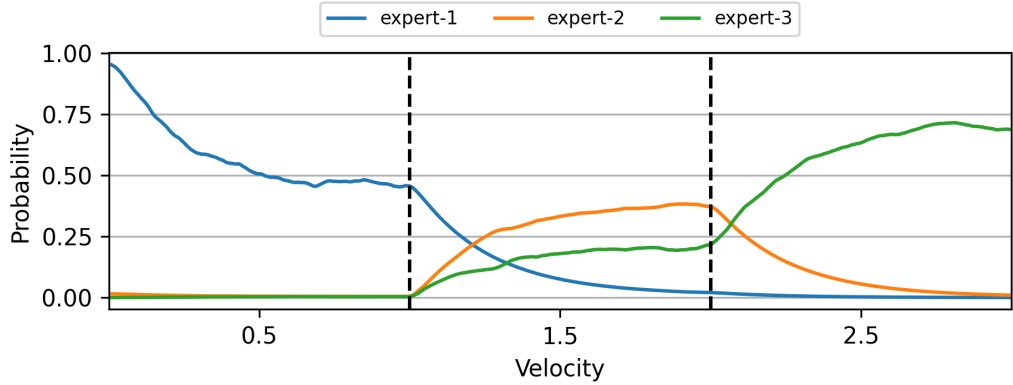

Figure 13: Probability of task assignments to some experts. The goal velocities are sampled from three distributions and divided by dashed lines.

Table 17: Quantification of the gradient conflict.

| Methods | Point-Robot | Cheetah-Vel | Walker-Param | Reach | Push |
|---------|-------------|-------------|--------------|-------|------|
| AD | 31.1% | 33.9% | 23.3% | 20.2% | 12.8% |
| T2MIR-AD | 7.4% | 1.2% | 1.1% | 15.6% | 1.8% |

## L Limitations

We discuss the limitations and computational efficiency of our method in this section. Constrained by limited computing resources, our method is trained on lightweight datasets, e.g., the Mixed datasets of Push contain 500 episodes per task and 45 training tasks, which have totally 2.25 million transitions. Although the size of datasets are small, we find it's enough to gain a high performance. But the size of datasets limits the capacity of MoE in extending the size of model parameters. Future work may evaluate their study on more complex environments and larger datasets such as XLand-MiniGrid [57, 10]. The efficiency of contrastive loss in task-wise MoE when facing massive number of tasks is not thoroughly explored in this work. It is interesting to explore whether the contrastive loss is effective for managing task assignments in scenarios involving a large number of tasks. Integrating the T2MIR framework incurs a slightly higher computational cost during the training process, but it requires less computational resources during task inference.

