# OpenReview forum: "Mixture-of-Experts Meets In-Context Reinforcement Learning"
_NeurIPS.cc/2025/Conference — NeurIPS 2025 poster_

### Official Review · Reviewer_ptNc · 2025-06-23

**Clarity:** 2
**Significance:** 2
**Originality:** 2
**Rating:** 4
**Confidence:** 4

**Summary:**

The paper tackles the challenge of in-context reinforcement learning (ICRL). To cope with (i) multi-modality of state-action-reward tokens and (ii) the wide variety of decision tasks, the authors replace each transformer feed-forward block with two Mixture-of-Experts (MoE) layers: one that routes individual tokens by modality, and another that routes whole trajectories by task. Evaluated on four benchmark suites—DarkRoom, Point-Robot, MuJoCo, and Meta-World—the proposed architecture surpasses ICRL and offline meta-RL baselines.

**Questions:**

1. Do you have quantitative evidence of gradient conflict to justify the task-wise MoE?
2. Can you isolate the modality issue so as to substantiate that semantic discrepancy is a major challenge of ICRL?
3. Have you evaluated T2MIR on more challenging benchmarks?

**Ethical Concerns:**

["NO or VERY MINOR ethics concerns only"]

**Final Justification:**

The authors have adequately addressed my concerns regarding empirical evidence and benchmarks. I have raised my score accordingly.

**Limitations:**

yes

**Paper Formatting Concerns:**

I haven’t noticed any major formatting issues.

**Quality:**

2

**Strengths And Weaknesses:**

**Strengths**
1. The method outperforms competitive ICRL and offline meta-RL baselines across DarkRoom, Point-Robot, multiple MuJoCo locomotion tasks, and Meta-World Reach & Push.
2. The paper carefully derives the dual MoE routing scheme and its contrastive objective, making the architectural choices easy to follow.
3. The ablation study and t-SNE visualizations shed light on how token- and task-level experts contribute to the gains.

**Weaknesses**
1. The paper asserts “the learning efficiency can be impeded by intrinsic gradient conflicts in challenging scenarios with significant task variation,” yet provides no empirical evidence of such conflicts in the pre-training phase of ICRL. Since the online adaptation phase is gradient-free, the relevance of this issue remains unclear.
2. While the authors motivate token-wise MoE by the heterogeneity of state, action and reward tokens, they do not quantify how serious this mismatch is for existing ICRL backbones.
3. Experiments focus on relatively simple tasks (e.g., Meta-World Reach/Push). It is uncertain whether the approach scales to harder suites such as the full ML1 or ML10.

---

> ### Author Rebuttal · Authors · 2025-07-30
>
> ***
>
> **`Q1. Empirical evidence of gradient conflicts in ICRL pre-training.`**
>
> A1. In the original manuscript, we have visualized the comparison of gradient conflicts between T2MIR-AD (MoE) and AD (MLP) as shown in Fig. 7 (L297, Page 9). It shows that **AD suffers from significant gradient conflict between opposing tasks and fails to discriminate tasks**. In contrast, T2MIR-AD maintains nearly orthogonal gradients across diverse tasks, especially for opposing ones. This comparison verifies **T2MIR’s advantage in both gradient conflict mitigation and task discrimination.**
>
> Further, we also provide quantification of the gradient conflict issue. The following table presents the proportion of task pairs with negative correlations among all pairs of tasks, which demonstrates **T2MIR’s significant superiority in mitigating gradient conflicts.**
>
> | **Gradient Conflict** | Point-Robot | Cheetah-Vel | Walker-Param | Reach | Push |
> | --- | --- | --- | --- | --- | --- |
> | **AD** | 31.1% | 33.9% | 23.3% | 20.2% | 12.8% |
> | **T2MIR-AD** | **7.4%** | **1.2%** | **1.1%** | **15.6%** | **1.8%** |
>
> Thank you for your insightful comment. In addition to Section 5 Experiments, **we will include empirical evidence of gradient conflicts and corresponding analysis in Section 1 Introduction, to further justify our motivation for the task-wise MoE**.
>
> ***
>
> **`Q2. Isolate and quantify the multi-modality issue of state-action-reward tokens.`**
>
> A2. In the original manuscript, we can observe the performance gain when addressing the multi-modality issue by token-wise MoE in Section 5.2 Ablation Study (Figure 4, Page 8). The AD and DPT baselines obtain relatively poor performance, and **incorporating the token-wise MoE induces a 6$\sim$36% return increase for AD and a 2$\sim$12% increase for DPT**.
>
> Further, motivated by your insightful comment, **we completely isolate the multi-modality issue and try to quantify the challenge incurred by this semantic discrepancy**. We manually design an MoE structure where state-action-reward tokens are routed to three experts by a **heuristic** gating scheme, i.e., one expert for one modality. The following table shows that **the heuristic gating mechanism can improve ICRL performance by a 3$\sim$24% increase in received return**. It again demonstrates the significant challenge induced by the semantic discrepancy across multi-modal tokens in the state-action-reward sequence, and the significant superiority of our token-wise MoE design.
>
> | **Ablation** | **AD** | **T2MIR-AD (heuristic gate in token-wise MoE)** | **T2MIR-AD** |
> | --- | --- | --- | --- |
> | Cheetah-vel | −119.2 $\pm$ 30.4 | -91.1 $\pm$ 30.4 | **−68.9** $\pm$ 2.1 |
> | Walker-Param | 395.2 $\pm$ 7.1 | 407.1 $\pm$ 10.5 | **435.7** $\pm$ 7.2 |
>
> Thank you for your insightful comment. In addition to Section 5 Experiments, **we will include the above analysis in Section 1 Introduction, to further justify our motivation for the token-wise MoE**.
>
> ***
>
> **`Q3. Evaluate T2MIR on more challenging benchmarks such as the full ML1 or ML10.`**
>
> A3. Following the main practice in literature [1-3], we adopt various benchmarks that are popular to evaluate in-context RL algorithms. Following your advice, we have conducted new evaluations on the challenging Ant-Dir environment in MuJoCo and the ML10 environment in Meta-World to demonstrate whether T2MIR can scale to these harder suites. The following table shows **T2MIR’s promising scalability and consistent superiority over various baselines in these challenging benchmarks**.
>
> | **Method** | **UNICORN** | **IDT** | **DICP** | **DPT** | **AD** | **T2MIR-DPT** | **T2MIR-AD** |
> | --- | --- | --- | --- | --- | --- | --- | --- |
> | Ant-Dir | 450.9 $\pm$ 10.2 | 643.7 $\pm$ 34.5 | 690.3 $\pm$ 20.4 | 634.1 $\pm$ 46.7 | 432.2 $\pm$ 42.5 | **708.6** $\pm$ 24.5 | 681.5 $\pm$ 30.4 |
> | ML10 | 193.6 $\pm$ 10.7 | 223.3 $\pm$ 13.3 | 215.9 $\pm$ 11.2 | 230.6 $\pm$ 25.1 | 277.0 $\pm$ 39.0 | 291.3 $\pm$ 21.9 | **318.1** $\pm$ 11.3 |
>
>
>
> [1] In-context RL with algorithm distillation, ICLR 2023.
>
> [2] Supervised pretraining can learn in-context RL, NeurIPS 2023.
>
> [3] Distilling RL algorithms for in-context model-based planning, ICLR 2025.
>
> ***
> ### **`Summary Response`**
>
> Thank you for your valuable review comments, which help us gain more critical insights and further enhance our empirical evaluation. We are honored to have **your positive comments on our motivation** ("making the architectural choices easy to follow") **and empirical evaluation** ("the ablation study and t-SNE visualizations shed light on…").
>
> Also, we are grateful that **other reviewers** (we refer to wTXA as R1, yZc9 as R2, and sbb8 as R3) **make positive comments to our overall contributions, including the novelty and motivation** (”the insight of…is interesting” by R1, "the idea of…sounds promising" by R2, and "propose a novel architecture" by R3), **the empirical evaluation** (”results demonstrate faster convergence and better rewards”, by R1, "a good set of experiments addressing various aspects of the work, which I appreciate" by R2, and "the experiments…are comprehensive, the ablations and analysis are insightful" by R3), **and the writing** (”the paper is fairly well-written” by R2). We hope that our contributions can be adequately justified.
>
> We summarize that your main concerns may include the evidence of the two ICRL challenges and the evaluation on more challenging benchmarks. **We have extended a number of justifications and experimental analyses to address your concerns**. Please let us know if we have addressed your concerns. We are more than delighted to have further discussions and improve our manuscript. **If our response has addressed your concerns, we would be grateful if you could re-evaluate our work**.

---

> > ### Author Response · Authors · 2025-08-03
> > **Looking forward to further discussions!**
> >
> > Dear Reviewer,
> >
> > Thank you for your continued effort in reviewing our work. We were wondering if our response and revision have resolved your concerns. We have addressed your initial questions through our rebuttal and are eager to clarify any further points you might raise. Please feel free to provide additional feedback. We greatly appreciate your continued engagement.
> >
> > Best regards,
> >
> > Authors

---

> > ### Comment · Reviewer_ptNc · 2025-08-05
> >
> > Thank you to the authors for addressing my concerns about empirical evidence and benchmarks. I've updated my score and hope the revised version will reflect these improvements.

---

> > > ### Author Response · Authors · 2025-08-05
> > > **Thank you!**
> > >
> > > Thank you very much for your constructive feedback and your time in helping us improve our work. We sincerely appreciate your updating the score on our work! We will reflect these improvements about empirical evidence and extended benchmarks in our revised version.

---

### Official Review · Reviewer_sbb8 · 2025-07-01

**Clarity:** 3
**Significance:** 2
**Originality:** 2
**Rating:** 5
**Confidence:** 4

**Summary:**

This paper introduces a framework named T2MIR which embeds token-wise MoE and task-wise MoE into transformer layers to handle multimodal ICRL data and diverse tasks. The authors choose AD and DPT as backbone models of the framework and evaluated them on five multi-task benchmarks. Across mixed datasets of the five benchmarks, both T2MIR-AD and T2MIR-DPT outperform strong baselines in learning speed and final return.

**Questions:**

1. Clarification of “multi-modality” among states, actions, and rewards - In many environments, these signals are deeply intertwined. For instance, in Cheetah-Vel the state comprises each segment’s angles, velocities, and angular velocities; the action is the torques applied to those segments; and the reward is a deterministic function of the resulting state. Could the authors clarify in what sense such tightly coupled signals are treated as separate “modalities” in their token-wise MoE routing?
2. Evaluation under IID tasks but OOD offline data - Have the authors tested T2MIR when tasks are sampled IID from P(M) but the offline dataset is out-of-distribution—e.g., collected by a random policy or by a policy trained on a different task? Evaluating under this dataset shift would more rigorously demonstrate T2MIR’s robustness.

**Ethical Concerns:**

["NO or VERY MINOR ethics concerns only"]

**Final Justification:**

The extensive experiments have addressed my concerns well and the OOD results significantly extend the scope of the work, showing the robustness to OOD data in offline RL. I have updated my score

**Limitations:**

Yes

**Quality:**

3

**Strengths And Weaknesses:**

Strengths:
 - The authors propose a novel architecture to integrate MoE into ICRL problems to tackle multi-modality transition and task heterogeneity.
 - The experiments on five diverse benchmarks and varying data qualities are comprehensive, showing robust performance gains of T2MIR.
 - The ablations and analysis are inisghtful. Ablation studies isolate the contributions of two MoEs; the t-SNE andn gradient-cosine plots illustrate expert specialization and conflict mitigation.

Weakness:
 - The DarkRoom and Point-Robot benchmarks seem to be weak in modern RL research. I suggest the authors consider other Mujoco benchmarks like AntDir and Humanoid, or text based benchmarks like webshop [1].
 - In this paper, the task inference is based on InfoNCE which takes hidden states and past trajectories to infer a task embedding. UNICORN broke a transition tuple (s, a, r, s') into two components: (s, a) that is more related to behavior policy, and (r, s') that is more related to the task [2]. I suggest the authors analyze how each component contributes to the task representation in their work, e.g., given two transitions from two tasks that have the same (s,a), but different (r, s'), how their task representations differ. I believe this analysis is important to show the robustness of the work in OOD scenario.

[1] Yao, Shunyu, et al. "Webshop: Towards scalable real-world web interaction with grounded language agents." Advances in Neural Information Processing Systems 35 (2022): 20744-20757.
[2] Li, Lanqing, et al. "Towards an information theoretic framework of context-based offline meta-reinforcement learning." arXiv preprint arXiv:2402.02429 (2024).

---

> ### Author Rebuttal · Authors · 2025-07-30
>
> ***
>
> **`Q1. Evaluation on other benchmarks like Ant-Dir and Humanoid, or text-based benchmarks like webshop.`**
>
> A1. Following the main practice in literature [1-3], we adopt various benchmarks that are popular to evaluate in-context RL algorithms. Following your advice, **we have conducted new evaluations on Ant-Dir in the MuJoCo benchmark and ML10 in the Meta-World benchmark** to demonstrate whether T2MIR can scale to these harder suites. The following table shows **T2MIR’s promising scalability and consistent superiority over various baselines**.
>
> | **Method** | **UNICORN** | **IDT** | **DICP** | **DPT** | **AD** | **T2MIR-DPT** | **T2MIR-AD** |
> | --- | --- | --- | --- | --- | --- | --- | --- |
> | Ant-Dir | 450.9 $\pm$ 10.2 | 643.7 $\pm$ 34.5 | 690.3 $\pm$ 20.4 | 634.1 $\pm$ 46.7 | 432.2 $\pm$ 42.5 | **708.6** $\pm$ 24.5 | 681.5 $\pm$ 30.4 |
> | ML10 | 193.6 $\pm$ 10.7 | 223.3 $\pm$ 13.3 | 215.9 $\pm$ 11.2 | 230.6 $\pm$ 25.1 | 277.0 $\pm$ 39.0 | 291.3 $\pm$ 21.9 | **318.1** $\pm$ 11.3 |
>
> [1] In-context RL with algorithm distillation, ICLR 2023.
>
> [2] Supervised pretraining can learn in-context RL, NeurIPS 2023.
>
> [3] Distilling RL algorithms for in-context model-based planning, ICLR 2025.
>
> ***
>
> **`Q2. Analysis of how the behavior policy and task-related information contribute to the task representation, respectively.`**
>
> A2. In our settings, we generate datasets by employing various checkpoints from different training steps, where one checkpoint (i.e., one behavior policy) may collect several trajectories. In order to extract the high-level representation from the gating output in task-wise MoE, we utilize contrastive learning to pull the trajectory representations from the same task closer while pushing those from different tasks farther apart. Intuitively, as the trajectories from the same task may be collected by different behavior policies, **pulling these representations closer means forcing the representations to exclude information of behavior policy and focus on task-related information in the trajectory**.
>
> Further, **we also conduct new experiments to assess T2MIR’s capacity to handle OOD offline data**, with the results presented in A4 to Q4. Specifically, we employ a random policy to interact with the environment and collect one trajectory to serve as a prompt for both T2MIR-DPT and DPT. Despite the significantly low return of the random trajectory, T2MIR-DPT still gains impressive improvement than DPT.
>
> ***
>
> **`Q3. Clarify the different modalities within the state-action-reward sequence in token-wise MoE.`**
>
> A3. As stated in the paper (L139-142), **the physical information contained within the state-action-reward sequence is different**. The state is typically continuous in nature, indicating that $s_{t+1}$ evolves based on $s_t$,  and contains some physical quantities of the agent (e.g., position, velocity, and acceleration). The action $a_t$ is less smooth (i.e., $a_{t+1}$ may be very different from $a_t$, although $s_{t+1}$ is similar to $s_t$), and may contain some mechanical control quantities, such as joint torques. The reward is a simple scalar that depends on state and action. Consequently, **there exists a substantial semantic gap among these three elements in the sequence**. Hence, we consider these three elements as different modalities and introduce a token-wise MoE to adaptively handle their distinct semantics. We will provide clearer clarifications on these separate modalities.
>
> ***
>
> **`Q4. Demonstrate T2MIR’s robustness by evaluating on OOD offline data.`**
>
> A4. In our experiments, we evaluate T2MIR and baselines via direct online interaction with the test environment, without access to any offline data. That is, the prompts or contexts are generated from online environment interactions to infer task beliefs during testing. Following your suggestion, **we have conducted new experiments to evaluate T2MIR’s robustness to OOD offline data as the test prompt**. Similar to the DPT setting where offline datasets are used as prompts, we employ a random behavior policy to interact with the environment and collect one trajectory as the offline prompt. The table below demonstrates **T2MIR’s consistent superiority even when prompted with OOD offline data**.
>
> | **Methods** | Point-Robot | Cheetah-Vel | Walker-Param |
> | --- | --- | --- | --- |
> | **DPT** | -5.9 $\pm$ 0.1 | -63.2 $\pm$ 4.4 | 419.0 $\pm$ 11.5 |
> | **T2MIR-DPT** | **-5.0** $\pm$ 0.2 | **-35.0** $\pm$ 2.9 | **424.1** $\pm$ 17.3 |
>
> Furthermore, **we conduct new experiments to evaluate T2MIR’s capability to handle OOD test tasks** in Cheetah-Vel. The target velocities in training tasks are uniformly sampled from $U[0.075,3]$, but during testing we sample OOD tasks from $U[3.1,3.3]$. The result presented below demonstrates **T2MIR’s consistent superiority even when dealing with OOD test tasks**.
>
> | **OOD Testing** | **UNICORN** | **IDT** | **DICP** | **DPT** | **AD** | **T2MIR-DPT** | **T2MIR-AD** |
> | --- | --- | --- | --- | --- | --- | --- | --- |
> | Cheetah-Vel  | -88.6 $\pm$ 2.1 | -114.3 $\pm$ 2.3 | -109.1 $\pm$ 2.1 | -78.5 $\pm$ 2.6 | -158.8 $\pm$ 47.1 | **-71.0** $\pm$ 0.7 | -117.3 $\pm$ 4.4 |
>
> ***
>
> ### **`Summary Response`**
>
> Thank you for your valuable review comments, which help us gain more critical insights and further enhance our empirical evaluation. We are honored to have **your recognition on our method, especially on its novelty** ("propose a novel architecture ") **and empirical evaluation** ("the experiments…are comprehensive, the ablations and analysis are insightful").
>
> Also, we are grateful that **other reviewers** (we refer to wTXA as R1, yZc9 as R2, ptNc as R4) **make positive comments to our overall contributions, including the novelty and motivation** ("the insight…is interesting" by R1, "the idea of…sounds promising" by R2, and "making the architectural choices easy to follow" by R4), **the empirical evaluation** ("results demonstrate faster convergence and better rewards" by R1, "a good set of experiments addressing various aspects of the work, which I appreciate" by R2, and "the ablation study and t-SNE visualizations shed light on…" by R4), **and the writing** (”the paper is fairly well-written” by R2). We hope that our contributions can be adequately justified.
>
> We summarize that your main concerns may include: 1) the evaluation on harder benchmarks, 2) deeper analysis on task representation learning, and 3) the robustness to OOD testing. **We have extended a number of justifications and experimental analyses to address your concerns**. Please let us know if we have addressed your concerns. We are more than delighted to have further discussions and improve our manuscript. **If our response has addressed your concerns, we would be grateful if you could re-evaluate our work**.

---

> > ### Comment · Reviewer_sbb8 · 2025-08-05
> >
> > Great thanks to the authors' comprehensive response! These experiment results, especially OOD experiments strongly support the contribution of the work.

---

> > > ### Author Response · Authors · 2025-08-06
> > > **Thank you!**
> > >
> > > Thank you for your continued effort in reviewing our work. We are honored to have your recognition on our work and response. Please feel free to provide additional feedback, and we are looking forward to any further discussions. We sincerely appreciate your continued engagement.

---

### Official Review · Reviewer_yZc9 · 2025-07-02

**Clarity:** 2
**Significance:** 3
**Originality:** 2
**Rating:** 4
**Confidence:** 3

**Summary:**

This work addresses the challenge of designing a scalable in-context RL framework capable of handling multimodality and diverse task distributions within a single transformer model. To this end, it introduces a Mixture of Experts (MoE) architecture that augments the transformer’s ability to manage the complexity arising from multimodality induced by states, actions, and rewards, as well as the generalization difficulties posed by task diversity. Specifically, the proposed framework replaces the standard feedforward layers in causal transformer blocks with two specialized modules: a token-wise MoE and a task-wise MoE. Experimental results across a range of tasks demonstrate the effectiveness of these modifications, and ablation studies further validate the design choices.

**Questions:**

1. In the Figure 6 analysis, when the token-wise MoE count is changed, how is the task-wise MoE count handled? I think they could be related.
2. Also, in the Figure 6 analysis, how is the decision to activate one-third of experts in token-wise MoE and always 2 experts in task-wise MoE reached?
3. Since the task variations seem to be parameterized in the experiments, do you also use the task variation parameterization in the context?

**Ethical Concerns:**

["NO or VERY MINOR ethics concerns only"]

**Final Justification:**

The proposed Mixture of Experts-based architecture seems promising, and the authors have done a good set of experiments to show the value of the method. While the techniques used in this work are not necessarily novel, putting them together in use for in-context RL is interesting. The rebuttal was responsive to all my major points, and I think this work is of value to the community. Therefore, I lean towards an accepting score, considering the novelty of the idea, the techniques used, the rigour of experiments conducted, and the final evaluation results.

**Limitations:**

Yes. It is given in the appendix.

**Paper Formatting Concerns:**

I do not have any concerns about the formatting.

**Quality:**

2

**Strengths And Weaknesses:**

The idea of using the MoE to tackle the multimodality and generalization challenges sounds promising, and as the paper has shown, the empirical evidence suggests that it has the potential to improve the performance of in-context RL. The authors have conducted a good set of experiments addressing various aspects of the work, which I appreciate. The paper is fairly well-written, though improvements can be made to improve the comprehension.

On a constructive note, considering the central role of task distributions in this work, the definition of task variation remains somewhat ambiguous. Appendix D.1 implies that these variations may be parameterized, in which case it would be natural to formalize the problem within the framework of a Contextual Markov Decision Process (CMDP). However, if the task variations are not consistently parameterizable, a different formalization may be more suitable. As currently presented, the paper lacks a clear formalism and specification of how tasks differ, making it challenging to interpret the relationships between tasks or understand how their difficulty evolves, specifically in interpreting the results.

Second, upon examining Figure 1 and the ablation in Section 5.4 on token-wise and task-wise MoE visualizations, I began to question whether learning the gating mechanism in MoEs is actually necessary. For instance, in the left figure of Figure 1—if I'm interpreting it correctly—it seems that three experts specializing in state, action, and reward tokens may suffice to get reasonably competing performance. This raises the possibility of manually defining a fixed gating scheme based on token type. Similarly, in the right panel of Figure 2, a simple heuristic gating based on evenly splitting the velocity range could be applied (this is also why I think the formalism on how task variations should be defined, as it tells you if one could do this or not). I'm curious how such heuristic approaches would perform compared to learned gating. Such an approach would not require separate loss terms like balancing and contrastive loss. It’s possible that I may be overlooking something, as the authors have likely considered this aspect carefully, so I would appreciate the authors' thoughts on the matter.

The experiments conducted are nice and cover the major important axes of interest. However, it would have been nice to have ablations to show the impact of balancing loss and contrastive loss. The authors provide the intuition as to why they are needed, and I agree with them, but it would be nice to see the impact scale of the proposed loss terms. Similarly, I could imagine challenges of training with the contrastive loss when the number of tasks is large. Any thoughts/analysis that authors have on this end?

---

> ### Author Rebuttal · Authors · 2025-07-29
>
> ***
>
> **`Q1. Clear formalism and specification of task variation, and the use of task variation parameterization in the context.`**
>
> A1. We follow the general setting in in-context RL domains, where **tasks are sampled from a distribution but task variations cannot be parameterized**. This is a more realistic setting for practical applications, as we cannot access some oracle information about the underlying task (i.e., the goal position in a navigation environment). Appendix D.1 introduces the details about how to construct the multi-task environments and corresponding datasets. Taking Cheetah-Vel as an example, tasks differ in the goal velocity, and we uniformly sample the goal velocity from the distribution $U[0.075, 3.0]$ to construct the multi-task dataset. In Figure 1, we visualize the gating representations of tasks with different goal velocities to gain deep insights into the relationship between the learned gating mechanism and task similarity. But, **we cannot access the oracle task information (i.e., goal velocity) during in-context training**. Following your advice, we will make a clear formalism on task variation.
>
> ***
>
> **`Q2. The necessity of learning the gating mechanism in MoEs, and its superiority over heuristic gating approaches.`**
>
> A2. As mentioned in our A1 to Q1, since we cannot access the oracle task information, it is infeasible to manually route the task-wise MoE based on heuristics (e.g., splitting the goal velocity range). In the token-wise MoE, we can manually assign tokens of three modalities to three experts, i.e., one expert for one modality. We have conducted additional ablations to gain insights into the heuristic gating mechanism. The result in the following table shows **the significant superiority of our gating mechanism over heuristic gating in the token-wise MoE**.  We conjecture that the improvement can come from the flexibility and adaptability of our end-to-end gating mechanism.
>
> | **Ablations** | **AD** | **T2MIR-AD (heuristic gate in token-wise MoE)** | **T2MIR-AD** |
> | --- | --- | --- | --- |
> | Cheetah-vel | −119.2 $\pm$ 30.4 | -91.1 $\pm$ 30.4 | **−68.9** $\pm$ 2.1 |
> | Walker-Param | 395.2 $\pm$ 7.1 | 407.1 $\pm$ 10.5 | **435.7** $\pm$ 7.2 |
>
> Further, we are striving to **design more universal, end-to-end architectures, with minimal requirements on domain knowledge or manual interventions**. This principle aligns with **Sutton’s The Bitter Lesson**: While task-specific designs may offer immediate gains, scalable approaches provide more sustainable long-term benefits.
>
> ***
>
> **`Q3. Ablations to show the impact of balancing loss and contrastive loss.`**
>
> A3. Following your suggestion, we have conducted new ablation studies to investigate the respective impacts of the two loss terms. We compare T2MIR to three new ablations:
>
> - **w/o balance_loss**: omitting balance loss (Eq.3) in token-wise MoE, while retaining contrastive loss in task-wise MoE.
> - **w/o contrastive_loss**: omitting contrastive loss (Eq.7) in task-wise MoE, while retaining balance loss in token-wise MoE.
> - **w/o aux_loss**: omitting both balance loss and contrastive loss.
>
> For all ablations, we keep all other hyperparameters the same as in the main experiments. The result in the following table shows that **both the balance loss and contrastive loss play significant roles in T2MIR’s superior performance**. Ablating the balance loss will cause a bit more performance degradation than ablating the contrastive one.
>
> | **Ablation** | **w/o aux_loss** | **w/o balance_loss** | **w/o contrastive_loss** | **T2MIR** |
> | --- | --- | --- | --- | --- |
> | Cheetah-Vel | -112.4 $\pm$ 30.4 | -92.8 $\pm$ 1.6 | -76.7 $\pm$ 3.5 | **-68.9** $\pm$ 2.1 |
> | Walker-Param | 393.2 $\pm$ 6.7 | 416.0 $\pm$ 5.3 | 417.0 $\pm$ 2.6 | **435.7** $\pm$ 7.2 |
>
> To further investigate the impact, we sample a batch of data and record the top-2 expert selections at the last training step. The percentages of experts being selected are shown in the following table. When ablating balance loss, 83% tokens activate experts 1 and 2; when ablating contrastive loss, all tokens activate experts 1 and 2. The result shows **the homogeneous expert assignments without these losses**. In contrast, in Figure 1, we can observe that **T2MIR’s expert assignments are much more heterogeneous**, demonstrating **the effectiveness of the balance and contrastive losses for MoE training**.
>
> | **Ablations** | **w/o balance_loss** | **w/o contrastive_loss** |
> | --- | --- | --- |
> | Rank 1 | (1,2): 83% | (1,2): 100% |
> | Rank 2 | (1,3): 5.6% | 0% |
> | Rank 3 | (2,4): 5.1% | 0% |
>
> ***
>
> **`Q4. Training the contrastive loss when the number of tasks is large.`**
>
> A4. According to the literature, **a key ingredient in the success of many contrastive learning methods is using a larger batch size with sufficient negative samples**. Only when the batch size is big enough, the loss function can cover a diverse enough collection of negative samples, challenging enough for the model to learn meaningful representation to distinguish different examples.
>
> To gain more insights into the contrastive loss in T2MIR, we omit the token-wise MoE and investigate T2MIR’s performance with different numbers of training tasks (10, 20, and 45). The result in the following table shows that **T2MIR’s performance increases as the number of training tasks increases**. In the task-wise MoE, we treat samples from the same task as positive pairs and samples from other tasks as negatives. When the number of tasks increases, we can obtain a larger batch with more negative samples, contributing to more reliable contrastive learning.
>
> | **# training tasks** | **10-train/5-eval** | **20-train/5-eval** | **45-train/5-eval** |
> | --- | --- | --- | --- |
> | T2MIR (w/o Token-MoE) | -158.2 $\pm$ 25.5 | -120.3 $\pm$ 26.1 | −101.1 $\pm$ 29.7 |
>
> ***
>
> **`Q5. When the token-wise MoE count is changed, how is the task-wise MoE count handled?`**
>
> A5. The token-wise MoE and task-wise MoE are two parallel layers. When changing one side (e.g., the token-wise MoE count), we strictly keep the other side invariant (e.g., the task-wise MoE count or other hyperparameters) to investigate the robustness of each component.
>
> **`Q6. How is the decision to activate one-third of experts in token-wise MoE and always 2 experts in task-wise MoE reached?`**
>
> A6. Since there are three modalities of the state-action-reward tokens, we always activate one-third of experts for each specific token. For the simplest case, we have three experts and assign one expert to each modality. Hence, we will only activate one expert for routing each token. For task-wise MoE, we follow the common practice in popular works, such as Gshard [1], BASE layers [2], and MoE++[3], which usually use top-2 routing. We will provide clearer clarification about the MoE design.
>
> [1] GShard: Scaling Giant Models with Conditional Computation and Automatic Sharding, ICRL 2021.
>
> [2] BASE Layers: Simplifying Training of Large, Sparse Models, ICML 2021.
>
> [3] MoE++: Accelerating Mixture-of-Experts Methods with Zero-Computation Experts, ICLR 2025 Oral.
>
> ***
>
> ### **`Summary Response`**
>
> Thank you for your valuable review comments, which help us gain more critical insights and further enhance our empirical evaluation. We are honored to have **your recognition on our method,** especially on **its novelty** ("the idea of…sounds promising"), **empirical evaluation** ("a good set of experiments addressing various aspects of the work, which I appreciate"), and **the writing** (”the paper is fairly well-written”).
>
> Also, we are grateful that **other reviewers** (we refer to wTXA as R1, sbb8 as R3, ptNc as R4) **make positive comments to our overall contributions, including the novelty and motivation** ("the insight…is interesting" by R1, "propose a novel architecture" by R3, and "making the architectural choices easy to follow" by R4), **and the empirical evaluation** ("results demonstrate faster convergence and better rewards" by R1, "the experiments…are comprehensive, the ablations and analysis are insightful" by R3, and "the ablation study and t-SNE visualizations shed light on…" by R4). We hope that our contributions can be adequately justified.
>
> We summarize that your main concerns may include: 1) the task variation parameterization, 2) the necessity of learning the gating mechanism, and 3) the ablation on regularization losses. **We have extended a number of justifications and experimental analyses to address your concerns**. Please let us know if we have addressed your concerns. We are more than delighted to have further discussions and improve our manuscript. **If our response has addressed your concerns, we would be grateful if you could re-evaluate our work**.

---

> > ### Comment · Reviewer_yZc9 · 2025-08-07
> >
> > I thank the authors for their efforts in addressing my concerns and questions. I find their rebuttal to be responsive to all of my major points. I like the experiment done for Q2, and other additional ablations also support the authors' claims. I will keep my accepting score.

---

### Official Review · Reviewer_wTXA · 2025-07-03

**Clarity:** 3
**Significance:** 2
**Originality:** 3
**Rating:** 4
**Confidence:** 4

**Summary:**

The paper proposes improving in context RL performance by introducing explicit MoE layers that operate at token vs task level to ensure that transformers can represent the multi-modality due to state, actions and rewards efficiently. The paper demonstrates the efficacy of this architecture on environments like DarkRoom, MuJoCo, Meta-World and demonstrates faster convergence and better results than baseline methods.

**Questions:**

1. In the ablation experiments, its not very clear that when you remove MoE do you keep the parameter count of the model same as with MoE to ensure fair representation capacity?

**Ethical Concerns:**

["NO or VERY MINOR ethics concerns only"]

**Final Justification:**

Thanks to the authors for addressing the review comments and providing more empirical data! I have revised my ratings for the paper.

**Limitations:**

Yes

**Paper Formatting Concerns:**

Fig 2: the ordering in diagram is b,a,c and it’s a little hard to follow the text underneath. Suggest authors to redraw with a,b,c order.

**Quality:**

3

**Strengths And Weaknesses:**

Strengths:
1. The paper addresses the fact that introducing MoE leads to imbalance and tries to address them via explicit losses to ensure all experts can be utilized.
2. The insight of pooling trajectory to ensure interference of gradient is reduced is interesting.
3. Results demonstrate faster convergence and better rewards compared to baseline methods.

Weakness:
1. Adding two MoE blocks per layer adds a lot of complexity for both training and inference. Balancing the experts requires explicit loss, introduces even more hyper parameters and I am not completely convinced that additional complexity via explicit architectural bias for tasks such as in context RL is required.
2. The paper performs experiments on small sized data with relatively small modes and I am not completely sure if this scales well and if standard large models even require explicit architectural bias.

---

> ### Author Rebuttal · Authors · 2025-07-30
>
> ***
>
> **`Q1. The necessity of leveraging explicit architectural bias for in-context RL.`**
>
> A1. In the era of large language models (LLMs), **MoE is a promising architecture for managing computational costs when scaling up model parameters**, such as in Gemini, Llama-MoE, and DeepSeekMoE. The architectural advancement also extends to various domains such as computer vision and image generation. Particularly, recent studies [1,2] show that incorporating MoE leads to substantial performance increases for DRL, **providing strong empirical evidence towards developing scaling laws (a model’s performance scales proportionally to its size) for RL**.
>
> The success of LLMs comes with training huge models on a diversity of datasets, exhibiting the power and generalization of in-context learning. In-context RL aims to harness this potential to improve RL’s generalization to downstream tasks. Compared to single-task RL, **it is more urgent for in-context RL to leverage explicit architecture bias to establish possible scaling laws that enable training of large, generalizable foundation models**.
>
> Further, in our A3 to Q3, we demonstrate that adding two MoE blocks per layer yields $11\sim 42$% performance increase at the cost of $10\sim 14$% running time increase, with the same number of total activated parameters as baseline models. In our A4 to Q4, we show that T2MIR is insensitive to hyperparameters related to MoE layers, incurring a minimum cost for hyperparameter tuning. In summary, we hope that **the necessity and superiority of leveraging architectural bias for in-context RL could be sufficiently justified**.
>
> [1] Mixtures-of-experts unlock parameter scaling for deep RL, ICML 2024.
>
> [2] Don't flatten, tokenize! Unlocking the key to SoftMoE's efficacy in deep RL, ICLR 2025.
>
> ***
>
> **`Q2. The scalability of our method, with experiments on small-sized data with relatively small models.`**
>
> A2. Following the common approach in literature to evaluate model scalability [1-3], we evaluate the scalability of T2MIR-AD, i.e., the performance with respect to the model size and the dataset size. The following table shows that **introducing MoE while increasing model/dataset size can unlock further scaling potential of T2MIR**.
>
> | T2MIR-AD | 1.8M transitions | 3.6M transitions | 5.4M transitions |
> | --- | --- | --- | --- |
> | **4.8M parameters** | −68.9 $\pm$ 2.1 | -65.3 $\pm$ 8.6 | -54.8 $\pm$ 0.9 |
> | **5.8M parameters** | -69.0 $\pm$ 1.6 | -58.6 $\pm$ 0.8 | -51.1 $\pm$ 0.6 |
> | **7.2M parameters** | -61.2 $\pm$ 1.0 | -57.3 $\pm$ 0.7 | -49.6 $\pm$ 0.7 |
>
> Further, we have conducted new evaluations on the challenging Ant-Dir in MuJoCo and ML10 in Meta-World to demonstrate that our method can scale to these harder suites. The following table shows **T2MIR’s promising scalability and consistent superiority over various baselines**.
>
> | **Method** | **UNICORN** | **IDT** | **DICP** | **DPT** | **AD** | **T2MIR-DPT** | **T2MIR-AD** |
> | --- | --- | --- | --- | --- | --- | --- | --- |
> | Ant-Dir | 450.9 $\pm$ 10.2 | 643.7 $\pm$ 34.5 | 690.3 $\pm$ 20.4 | 634.1 $\pm$ 46.7 | 432.2 $\pm$ 42.5 | **708.6** $\pm$ 24.5 | 681.5 $\pm$ 30.4 |
> | ML10 | 193.6 $\pm$ 10.7 | 223.3 $\pm$ 13.3 | 215.9 $\pm$ 11.2 | 230.6 $\pm$ 25.1 | 277.0 $\pm$ 39.0 | 291.3 $\pm$ 21.9 | **318.1** $\pm$ 11.3 |
>
>
>
> Following the main practice in literature [4-6], we adopt various benchmarks that are popular to evaluate in-context RL algorithms. Further, as stated in our limitations and future work, our method is evaluated on widely adopted benchmarks in ICRL, with relatively lightweight datasets compared to popular large models. An urgent improvement is to evaluate on more complex environments such as XLand-MiniGrid and XLand-100B with huge datasets, unlocking the scaling properties of MoE in ICRL domains.
>
> [3] Network Sparsity Unlocks the Scaling Potential of Deep RL, ICML 2025.
>
> [4] In-context RL with algorithm distillation, ICLR 2023.
>
> [5] Supervised pretraining can learn in-context RL, NeurIPS 2023.
>
> [6] Distilling RL algorithms for in-context model-based planning, ICLR 2025.
>
> ***
>
> **`Q3. Complexity of adding two MoE blocks per layer for both training and inference.`**
>
> A3. For the two MoE blocks, we keep the total activated parameters the same as those in a normal feed-forward network (FFN) block. During training, only the activated experts need to compute gradients, so the cost is approximately the same as FFN. During inference, the additional cost of MoE mainly includes computing the expert distribution by the router. The following table shows that **introducing the MoE architecture only incurs a 10$\sim$14% increase in training time and a 14$\sim$27% increase in inference time, which is minor.**
>
> | Training/Inference Time | T2MIR-AD (MoE) | AD (FFN) | T2MIR-DPT (MoE) | DPT (FFN) |
> | --- | --- | --- | --- | --- |
> | Cheetah-Vel | 4h18min / 78.0s | 3h57min / 67.9s | 2h46min / 59.6s | 2h27min / 46.8s |
> | Walker-Param | 4h22min / 79.8s | 3h53min / 69.7s | 2h47min / 61.8s | 2h26min / 48.5s |
>
> ***
>
> **`Q4. Hyperparameters and additional loss for balancing the experts.`**
>
> A4. We have conducted hyperparameter analysis on MoE configurations in Section 5.3 and Appendix G.1. The results show that **a moderate configuration is more suitable for token-wise MoE, and the performance slightly increases with more experts in task-wise MoE**.
>
> Following your suggestion, we have conducted new ablation studies to investigate the respective impacts of additional losses for balancing the experts. We compare T2MIR to three new ablations:
>
> - **w/o balance_loss**: omitting balance loss (Eq.3) in token-wise MoE, while retaining contrastive loss in task-wise MoE.
> - **w/o contrastive_loss**: omitting contrastive loss (Eq.7) in task-wise MoE, while retaining balance loss in token-wise MoE.
> - **w/o aux_loss**: omitting both balance loss and contrastive loss.
>
> For all ablations, we keep all other hyperparameters the same as in the main experiments. The result in the following table shows that **both the balance loss and contrastive loss play significant roles in T2MIR’s superior performance**. Ablating the balance loss will cause a bit more performance degradation than ablating the contrastive one.
>
> | **Ablation** | **w/o aux_loss** | **w/o balance_loss** | **w/o contrastive_loss** | **T2MIR** |
> | --- | --- | --- | --- | --- |
> | Cheetah-Vel | -112.4 $\pm$ 30.4 | -92.8 $\pm$ 1.6 | -76.7 $\pm$ 3.5 | **-68.9** $\pm$ 2.1 |
> | Walker-Param | 393.2 $\pm$ 6.7 | 416.0 $\pm$ 5.3 | 417.0 $\pm$ 2.6 | **435.7** $\pm$ 7.2 |
>
> To further investigate the impact, we sample a batch of data and record the top-2 expert selections at the last training step. The percentages of experts being selected are shown in the following table. When ablating balance loss, 83% tokens activate experts 1 and 2; when ablating contrastive loss, all tokens activate experts 1 and 2. The result shows **the homogeneous expert assignments without these losses**. In contrast, in Figure 1, we can observe that **T2MIR’s expert assignments are much more heterogeneous**, demonstrating **the effectiveness of the balance and contrastive losses for MoE training**.
>
> | **Ablations** | **w/o balance_loss** | **w/o contrastive_loss** |
> | --- | --- | --- |
> | Rank 1 | (1,2): 83% | (1,2): 100% |
> | Rank 2 | (1,3): 5.6% | 0% |
> | Rank 3 | (2,4): 5.1% | 0% |
>
> ***
>
> **`Q5.  Maintaining fairness regarding the parameter count in ablation studies.`**
>
> A5. In ablation studies, we strictly keep the same number of activated parameters for the MoE and FFN architectures during training and inference, for a fair comparison.
>
> ***
>
> **`Q6. Formatting Concerns about Fig. 2.`**
>
> A6. Thanks a lot for your detailed suggestion, and we will improve the diagram and the text underneath.
>
> ***
>
> ### **`Summary Response`**
>
> Thank you for your valuable review comments, which help us gain more critical insights and further enhance our empirical evaluation. **We are honored to have your positive comments on our motivation** (”the insight of…is interesting”) **and the empirical evaluation** (”results demonstrate faster convergence and better rewards”).
>
> Also, we are grateful that **other reviewers** (we refer to yZc9 as R2, sbb8 as R3, ptNc as R4) **make positive comments to our overall contributions, including the novelty and motivation** ("the idea of…sounds promising" by R2, "propose a novel architecture" by R3, and "making the architectural choices easy to follow" by R4), **the empirical evaluation** ("a good set of experiments addressing various aspects of the work, which I appreciate" by R2, "the experiments…are comprehensive, the ablations and analysis are insightful" by R3, and "the ablation study and t-SNE visualizations shed light on…" by R4), **and the writing** (”the paper is fairly well-written” by R2). We hope that our contributions can be adequately justified.
>
> We summarize that your main concerns may include the necessity of leveraging architectural bias for ICRL, the model scalability, and the model complexity of our method. **We have extended a number of justifications and experimental analyses to address your concerns.** Please let us know if we have addressed your concerns. We are more than delighted to have further discussions and improve our manuscript. **If our response has addressed your concerns, we would be grateful if you could re-evaluate our work**.

---

> > ### Author Response · Authors · 2025-08-07
> > **Looking forward to further discussions!**
> >
> > Dear Reviewer wTXA,
> >
> > Thank you for your continued effort in reviewing our work. We were wondering **if our response and revision have resolved your concerns**. We have addressed your initial questions through our rebuttal and **are eager to clarify any further points you might raise**. Please feel free to provide additional feedback. We greatly appreciate your continued engagement.
> >
> > Best regards,
> >
> > Authors

---

> > > ### Comment · Reviewer_wTXA · 2025-08-08
> > >
> > > Thanks to the authors for addressing the review comments and providing more empirical data! I have revised my ratings for the paper.

---

> > > > ### Author Response · Authors · 2025-08-08
> > > > **Thank you!**
> > > >
> > > > Thank you very much for your constructive feedback and your time in helping us improve our work. We sincerely appreciate your revising the rating for our paper! Thanks a lot for your helpful comments, and we will include corresponding improvements and extended empirical evidence in our revised version.

---

### Note · Authors · 2025-08-13

Dear Reviewers, ACs, SACs, and PCs,

We sincerely appreciate your help and effort in offering a good atmosphere for the discussion phase, where we are experiencing a positive communication process. We thank all reviewers (we refer to wTXA as R1, yZc9 as R2, sbb8 as R3, and ptNc as R4) for providing valuable comments to improve our work and for providing timely feedback when we successfully addressed their concerns. We have made a number of changes to address reviewers' suggestions and concerns. A concise summary of the modifications is made as

- We conducted new experiments to **evaluate T2MIR’s scalability with various sizes of datasets and parameters**. (confirmed by R1, R2)
- We conducted new experiments in more challenging environments (e.g., Ant-Dir and ML10) to **show T2MIR’s consistent superiority against baselines**. (confirmed by R1, R3, R4)
- We showed the time cost during training and inference, demonstrating that **T2MIR yields a significant performance gain at the cost of a minor increase in running time**. (confirmed by R1)
- We conducted new ablation studies to **demonstrate the significance of both the balance loss and contrastive loss, and the effectiveness of our routing mechanism**. (confirmed by R1, R2, R4)
- We provided new numerical evidence of gradient conflicts for the FFN and MoE architectures to **further support the effectiveness of task-wise MoE**. (confirmed by R4)
- We conduct new experiments to **evaluate T2MIR’s generalization ability to out-of-distribution (OOD) offline data and test tasks**. (confirmed by R3)
- We provided more details about experiment settings and **clarified the promise of the MoE architecture for ICRL**. (confirmed by R1, R2, R3)
- We clarified the different modalities in token-wise MoE, and the impact of the behavior policy and task-related information in task-wise MoE. (confirmed by R2, R3, R4)

We are grateful that all reviewers made positive comments on our overall contributions, such as the novelty and motivation, the empirical evaluation, and the writing. Further, we are encouraged that **all reviewers gave positive feedback for our effective resolution of their concerns and expressed positive approval of our work**. We hope we were able to fully address all reviewers’ concerns to make our work more solid. Again, we truly appreciate your continued engagement in helping improve our work!

Best Regards,

The Authors

---

### Decision · Program_Chairs · 2025-09-17

**Decision:**

Accept (poster)

**Comment:**

This paper introduces a framework for in-context reinforcement learning that integrates Mixture-of-Experts (MoE) architecture into a transformer backbone. The core contribution is using a token-wise MoE to handle the distinct semantics of state, action, and reward data, and a parallel task-wise MoE to route diverse tasks to specialized experts. Extensive experiments demonstrate that the proposed architecture outperforms strong baselines.

Reviewers expressed concerns regarding the method's novelty and scalability, the added complexity of the architecture, and the empirical justification for its design. In particular about the initial evaluation, asking for tests on more challenging environments (wTXA, sbb8, ptNc), analysis of robustness to out-of-distribution data (sbb8), and stronger empirical evidence for the claimed benefits of the dual MoE structure (yZc9, ptNc). During the rebuttal many of these potential weaknesses were addressed including evidence showing the method scales to more difficult benchmarks (Ant-Dir, ML10), limited computational overhead, and justification for architectural choices.